# Multimodal binding and inhibition of bacterial ribosomes by the antimicrobial peptides Api137 and Api88

Simon M. Lauer [1,2,8], Maren Reepmeyer [3,4,8], Ole Berendes [5], Dorota Klepacki[6], Jakob Gasse [3,4], Sara Gabrielli [5], Helmut Grubmüller[5], Lars V. Bock [5], Andor Krizsan[3,4], Rainer Nikolay [1,7] ✉, Christian M. T. Spahn [1] ✉ & Ralf Hoffmann [3,4] ✉

Proline-rich antimicrobial peptides (PrAMPs) inhibit bacterial protein biosynthesis by binding to the polypeptide exit tunnel (PET) near the peptidyl transferase center. Api137, an optimized derivative of honeybee PrAMP apidaecin, inhibits protein expression by trapping release factors (RFs), which interact with stop codons on ribosomes to terminate translation. This study uses cryo-EM, functional assays and molecular dynamic (MD) simulations to show that Api137 additionally occupies a second binding site near the exit of the PET and can repress translation independently of RF-trapping. Api88, a C-terminally amidated (-CONH$_2$) analog of Api137 (-COOH), binds to the same sites, occupies a third binding pocket and interferes with the translation process presumably without RF-trapping. In conclusion, apidaecin-derived PrAMPs inhibit bacterial ribosomes by multimodal mechanisms caused by minor structural changes and thus represent a promising pool for drug development efforts.

The discovery of many antibiotics in the last century and their subsequent clinical use against pathogens has dramatically reduced human mortality and morbidity and enabled new clinical treatments, such as organ transplantation and cancer therapy. The widespread use of antibiotics has put high evolutionary pressure on pathogens to acquire bacterial resistance mechanisms, some of which existed already 30,000 years ago[1], through mutations or horizontal gene transfer between bacteria. This leads to treatment failures increasing the lethality of bacterial infections and, if not addressed, could cause more deaths from bacterial infections than cancer by 2050[2]. This pessimistic view stems mostly from seven species of the sentinel "ESKAPEE" pathogens of particular concern due to the rapid spread of multi- and pan-resistant strains, including *Escherichia coli*, accounting for more than 80% of the global deaths associated with antibiotic resistance[3]. Therefore, new antibiotics with novel mechanisms to overcome resistance mechanisms relevant for nosocomial infections need to be identified and further developed for clinical use.

Potential alternatives to small molecule antibiotics are gene-encoded antimicrobial peptides (AMPs), which are expressed in all kingdoms of life as part of innate immunity[4]. Pharmacologically, AMPs produced in higher organisms appear to be promising due to their

[1]Institute of Medical Physics and Biophysics, Charité – Berlin University of medicine, corporate member of Freie Universität Berlin and Humboldt Universität zu Berlin, Berlin, Germany. [2]Humboldt-Universität zu Berlin, Institut für Biologie, 10099 Berlin, Germany. [3]Institute of Bioanalytical Chemistry, Faculty of Chemistry and Mineralogy, Universität Leipzig, Leipzig, Germany. [4]Center for Biotechnology and Biomedicine, Universität Leipzig, Leipzig, Germany. [5]Theoretical and Computational Biophysics Department, Max Planck Institute for Multidisciplinary Sciences, Göttingen, Germany. [6]Department of Pharmaceutical Sciences, University of Illinois at Chicago, Chicago, IL 60607, USA. [7]Max Planck Institute for Molecular Genetics, Department of Genome Regulation, Ihnestrasse 63-73, 14195 Berlin, Germany. [8]These authors contributed equally: Simon M. Lauer, Maren Reepmeyer. ✉e-mail: nikolay@molgen.mpg.de; christian.spahn@charite.de; bioanaly@rz.uni-leipzig.de

presumed low cytotoxicity as host peptides. However, AMPs attacking bacterial membranes often have low safety margins, which does not allow them to be applied at pharmacologically desired high doses to maintain a minimum blood concentration for longer periods of time before the next injection. Therefore, such lytic peptides are typically considered for topical applications. AMPs that specifically inhibit intracellular bacterial targets are more promising lead compounds for systemic treatments. This includes insect-derived proline-rich AMPs (PrAMPs), which represent promising lead structures because humans, unlike other mammals, are unable to produce them as part of innate immunity and thus appear to have a gap in their antibacterial repertoire that could be filled therapeutically[5]. In addition, PrAMPs are inherently stable to proteases due to the high proline content, are non-toxic to mammalian cell lines, and well-tolerated in animals[5,6]. They penetrate the outer membrane of Gram-negative bacteria, use the bacterial transporter SbmA to enter the cytoplasm, and inhibit protein translation by interfering with the bacterial (70S) ribosome[7,8]. Mechanistically, PrAMPs can be divided into two classes based on their effect on the bacterial ribosome[5,8]. First, oncocin-type PrAMPs, such as Onc112, bind to the PET in reverse orientation compared to the nascent protein chain synthesized on the ribosome, thereby blocking the PET and stalling protein translation at the initiation of translation[9,10]. Second, Api137, a representative of apidaecin-type PrAMPs, binds to the PET of the large ribosomal subunit near the peptidyl transferase center (PTC) in the same orientation as the nascent protein chain. It inhibits translation termination by trapping the release factors RF1 and RF2, which arrest translating ribosomes at the stop codon or promote stop codon readthrough, resulting in the expression of C-terminally elongated proteins[11–13].

Here we have focused on two structurally modified versions of apidaecin 1b (GNNRPVYIPQPRPPHPRL), which was originally isolated from the western honeybee (*Apis mellifera*). Substitutions at positions 1 (Gly1Orn) and 10 (Gln10Arg) and N-terminal guanidation significantly improved the antibacterial activities of Api137 (Gu-ONNRP-VYIPRPRPPHPRL-OH; Gu: tetramethylguanidine; O: L-ornithine) and its C-terminally amidated analog Api88 (Gu-ONNRPVYIPRPRPPHPRL-NH₂)[14]. Api137 and Api88 differ only in their C-termini, i.e., a free acid versus an amide, which are negatively charged or neutral, respectively, at physiological conditions. The engineering of Api88 aimed at achieving greater stability by preventing C-terminal protease-mediated digestion, but unexpectedly resulted in higher cellular uptake rates[15,16]. Both PrAMPs are characterized by enhanced antibacterial activity against Gram-negative bacteria and demonstrated efficacy in various murine infection models using different routes of administration[17,18]. However, it is unclear whether and to what extent the molecular mechanism of action differs between Api137 and Api88, as the interaction of the C-terminal carboxyl group of Api137 with the P-site tRNA is suggested to be critical for the stable formation of the trapped RF-state[9,17].

By employing functional assays and cryo-EM structural investigations, we show that amidation of the C-terminus of Api137, yielding Api88, alters its mechanism of action. The neutral C-terminus of Api88 allows the molecule to move closer to the PTC, thereby shifting the binding site within the PET 3.2 Å further towards the subunit interface. In addition, the binding mode of Api88 appears more dynamic. Our cryo-EM density is not compatible with a single conformer as for Api137 but with at least three slightly different binding conformers of Api88 that most likely reduce entropic loss. The dynamic nature of Api88 binding is supported by molecular dynamics (MD) simulations initiated from the cryo-EM structures. Furthermore, an additional binding site on the solvent side of the PET was identified for both Api88 and Api137, representing a potential first attachment point on the ribosome during ongoing translation. Finally, a third binding site in domain III of the 50S subunit was found occupied only by Api88.

## Results

### Api137 and Api88 suppress ribosomal translation activity

Api137 binds to the ribosome and forms a complex with the RF and the ribosome, thereby interfering with protein translation. The mechanism of the closely related Api88 is thought to be very similar, despite significant differences in their in vitro properties. While their antibacterial activities are similar, and both share the ribosome as the main target, Api88 exhibits a significantly higher uptake rate, resulting in a faster accumulation of the peptide within the cell[7,15,16]. However, structural and biochemical studies highlighted the importance of the C-terminal carboxylate group in proper RF trapping[11,19]. This prompted us to investigate potential differences in the modes of action of Api137 and Api88. We first investigated their effect on protein translation in the presence or absence of the RFs using an in vitro transcription-translation (ITT) assay[20,21]. The production of superfolder GFP (sfGFP) in the absence of RF1 was reduced by both Api88 and Api137 to a similar extent, down to ~60% at the highest peptide concentration tested, 50 μmol/L (Fig. 1a–c). In the presence of RF1, the inhibitory effect of Api88 remained unaffected, whereas Api137 further reduced the sfGFP translation rate to only ~30% (Fig. 1a–c).

To further explore the translation inhibition of Api88, the effects of Api88 on ribosome progression were tested by toeprinting, which allows mapping of the site of inhibitor-induced ribosome stalling during in vitro translation[22]. Consistent with previous reports[11,21], Api137 arrested translation at the mRNA stop codon resulting in an intense toeprinting band at a concentration of 50 μmol/L (Fig. 1d). At this concentration, Api88 had no effect on translation initiation and resulted only in a faint toeprint band that was significantly weaker than the band observed for Api137, corresponding to ribosomes arrested at the UAG stop codon (Fig. 1d). This was consistent with the results of in vitro sfGFP translation (Fig. 1a–c), suggesting that Api88 does not arrest ribosomes about to release the nascent peptide chain at the stop codon.

### Api137 and Api88 bind ribosomes with different affinities

In addition to the bacterial ribosome as the main target, Api137 and Api88 also bind to other proteins, such as the heat shock protein DnaK[23] and ribosome-associated RFs[5,11,17]. These and other proteins present in previously used ribosome preparations[7,20,24] will most likely affect the measured dissociation constant ($K_d$). Therefore, we used an additional chromatographic purification to obtain highly pure *E. coli* 70S ribosomes (Supplementary Fig. 1)[25] and thus more reliable 70S ribosome binding constants. The ribosome binding was studied by a fluorescence polarization assay using 5(6)-carboxyfluorescein (cf)-labeled PrAMPs and pure 70S ribosomes as well as the pure large (50S) and small (30S) ribosomal subunits containing only traces of a few proteins. These contaminating proteins were detected by mass spectrometry but not by SDS-PAGE, which exhibited the expected pattern of ribosomal protein bands (Supplementary Fig. 1). The $K_d$ of cf-Api137 and purified 70S ribosome were ~23-fold higher (4.73 μmol/L) than previously reported for 70S ribosomal extracts (Fig. 1e), whereas the $K_d$ of Onc112 (VDKPPYLPRPRPPRrIYNr-NH₂, r: D-arginine) was very similar to previously reported data[7,20,24]. This suggests that contaminating proteins only affect Api137 binding (Fig. 1e). While the higher affinity for Api137 is likely attributed to the presence of RF1 (or other proteins) in crude 70S ribosome extracts, it was surprising to see that the $K_d$ of Api88 increased only slightly to 1.8 μmol/L (Fig. 1e), suggesting different binding behaviors or binding partners of Api137 and Api88. The binding of all three PrAMPs to the purified large (50S) and small (30S) subunits was further investigated (Fig. 1e). While the $K_d$ of Onc112 for the 50S subunit was similar to that of the fully assembled ribosome, the $K_d$-values of Api137 and Api88 were two- (2.2 μmol/L) and threefold (0.6 μmol/L) lower, respectively, indicating significantly stronger binding to the 50S subunit. Binding to the 30S subunit was approximately 23-fold weaker for Onc112 and Api88 and 10-fold

weaker for Api137 compared to the 50S subunit (Fig. 1e), suggesting that the 50S subunit is the major interaction partner of the PrAMPs studied. A competition assay examining the binding of cf-Api88 in the presence of unlabeled Api88, Api137, and Onc112 to the 70S ribosome revealed similar $K_i$-values ranging from 1.9 to 3.0 μmol/L (Supplementary Fig. 2), indicating that the major binding site of Api88 is located in the exit-tunnel of the 50S, partially overlapping with areas occupied by both Api137 and Onc112. This was further confirmed by erythromycin, which binds deeper in the exit tunnel of the 50S subunit, significantly competing with the binding of cf-Api88 (Supplementary Fig. 2), as previously shown for cf-Api137 and cf-Onc112[8].

Taken together, Api137 and Api88 have a comparable level of antimicrobial activity[18], but the inhibitory effect of Api88 on translating 70S ribosomes is largely independent of RF1 and ~50% weaker than for Api137 in the presence of RF1. Interestingly, Api88 binds to purified 70S ribosomes and 50S subunits with ~2-3-fold higher affinity than Api137. This indicates that the large ribosomal subunit is the primary target of Api88.

## UV-activatable Api88 forms cross-links with ribosomal proteins

Next, we tested whether Api88 interacts directly with ribosomal proteins. Thus, Tyr7 in Api88 was substituted with 4-benzoyl-L-phenylalanin, which allows UV-induced cross-linking to nearby residues of bound proteins. Additionally, the biotin-Ser-Gly moiety was attached to the side chain of Orn1 to yield biotin-SG-Api88(Y7B), which allows a specific enrichment of crosslinked proteins via streptavidin-coated magnetic beads for subsequent MS analysis. Previous studies showed similar minimum inhibitory activities and binding constants for

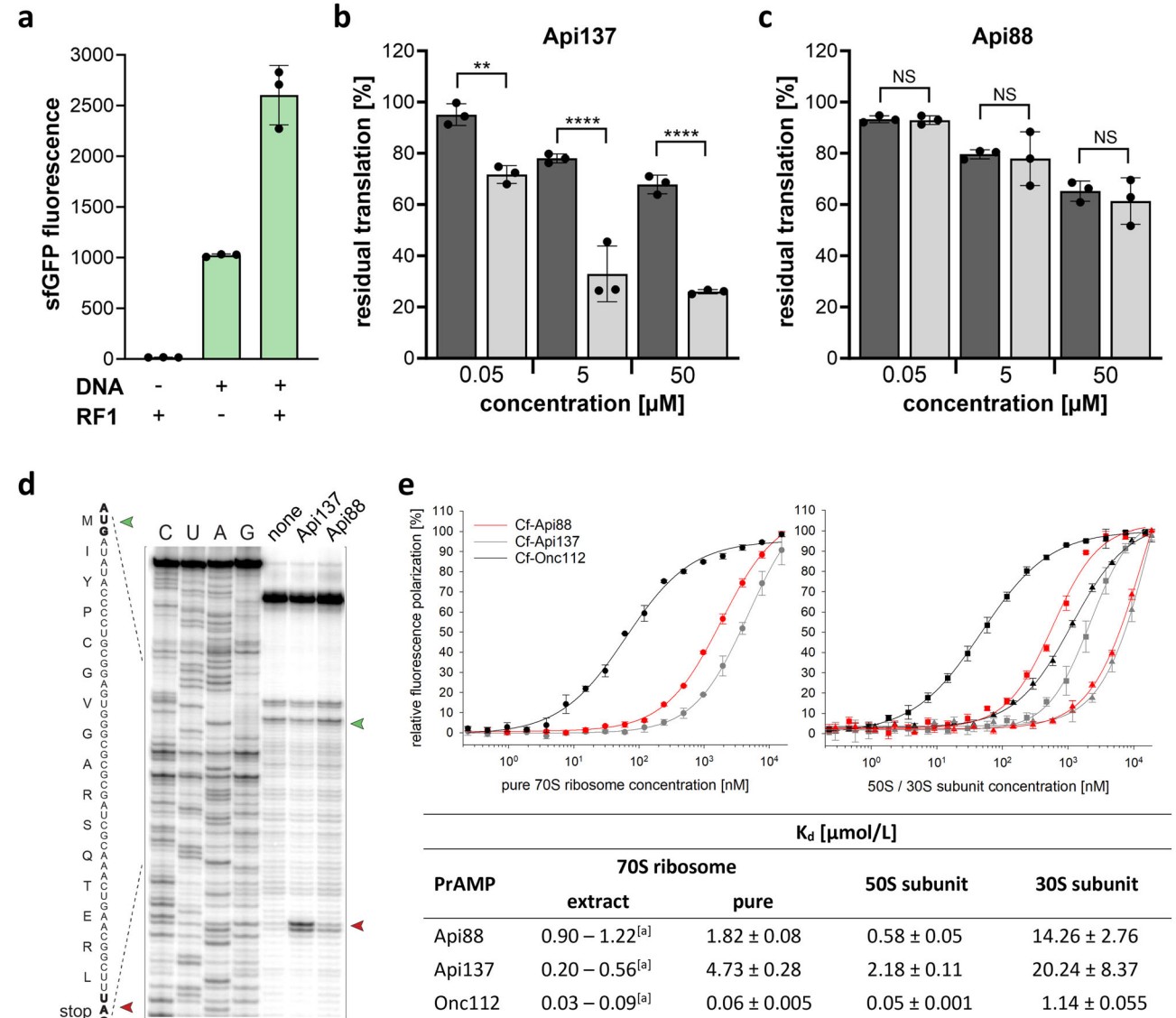

| Kd [μmol/L] | | | | |
|---|---|---|---|---|
| PrAMP | **70S ribosome** | | **50S subunit** | **30S subunit** |
| | extract | pure | | |
| Api88 | 0.90 − 1.22[a] | 1.82 ± 0.08 | 0.58 ± 0.05 | 14.26 ± 2.76 |
| Api137 | 0.20 − 0.56[a] | 4.73 ± 0.28 | 2.18 ± 0.11 | 20.24 ± 8.37 |
| Onc112 | 0.03 − 0.09[a] | 0.06 ± 0.005 | 0.05 ± 0.001 | 1.14 ± 0.055 |

**Fig. 1 | Inhibition of translation and ribosome binding by PrAMPs. a** In vitro expression of sfGFP, in the absence and presence of RF1 and inhibition of in vitro sfGFP translation by **b** Api137 and **c** Api88. Bar graphs in **b** and **c** represent the values of the sfGFP fluorescence using purified 70S ribosomes (lacking RF1) without (dark gray) or with RF1 supplementation (light gray), normalized to the fluorescence recorded in the absence of a PrAMP (set to 100%). Error bars indicate the deviation from the mean in three independent experiments (n = 3). Significance levels are indicated as NS (not significant), **(P value = 0.0016), and ****(P value < 0.0001) using one-way ANOVA with Tukey's multiple comparison test (GraphPad Prism). **d** In vitro toeprinting analysis of Api137 and Api88. The start codon is marked by a green and ribosome arrest at the UAG stop codon of the model *yrbA* ORF is marked by a red arrowhead. The control reaction with no added PrAMPs is labeled as "none". Sequencing reactions are labeled as C, U, A, G. The shown gel represents one of two replicates. **e** Binding of cf-labeled Api88 (red), Api137 (gray), and Onc112 (black) to purified 70S ribosome (circles, left), the 50S subunit (squares, right) and the 30S subunit (triangle, right) used to calculate dissociation constants (Kd, table) compared to previously reported Kd values using crude 70S ribosome extracts [a = [7,20,24]]. Error bars indicate the deviation from the mean of triplicates. Source data are provided as a Source Data file.

## Table 1 | Microscope data, refinement, and validation statistics

| | #1 Api137 EMDB-19426 PDB 8RPY | #2 Api88 conf. I EMD-19427 PDB 8RPZ | #3 Api88 conf. II EMD-19428 PDB 8RQ0 | #4 Api88 conf. III EMD-19429 PDB 8RQ2 |
|---|---|---|---|---|
| **Data collection and processing** | | | | |
| Magnification | 42,000x | 62,000x | 62,000x | 62,000x |
| Voltage (kV) | 300 | 300 | 300 | 300 |
| Electron exposure (e–/Å$^2$) | 24.6 | 26 | 26 | 26 |
| Defocus range (μm) | –0.5 to –2.0 | –0.5 to –2.0 | –0.5 to –2.0 | –0.5 to –2.0 |
| Pixel size (Å) | 1.03 (1.18) | 0.667 (0.998) | 0.667 (0.998) | 0.667 (0.998) |
| Symmetry imposed | C1 | C1 | C1 | C1 |
| Initial particle images (no.) | 626,116 | 353,243 | 353,243 | 353,243 |
| Final particle images (no.) | 424,855 | 260,475 | 260,475 | 260,475 |
| Map resolution (Å) | 2.64 | 2.44 | 2.44 | 2.44 |
| FSC threshold | 0.143 | 0.143 | 0.143 | 0.143 |
| Map resolution range (Å) | 2.643-34 | 2.2-31 | 2.2-31 | 2.2-31 |
| **Refinement** | | | | |
| Initial model used (PDB code) | 7K00[43] | 7K00[43] | 7K00[43] | 7K00[43] |
| Model resolution (Å) | 2.6 | 2.4 | 2.4 | 2.4 |
| FSC threshold | 0.143 | 0.143 | 0.143 | 0.143 |
| Model resolution range (Å) | 2.6-3.0 | 2.4-2.8 | 2.4-2.8 | 2.4-2.8 |
| Map sharpening $B$ factor (Å$^2$) | –78 | –60.8 | –60.8 | –60.8 |
| Model composition | | | | |
| Non-hydrogen atoms | 90,576 | 90,641 | 90,641 | 90,641 |
| Protein residues | 3,299 | 3,309 | 3,309 | 3,309 |
| Ligands | 0 | NH2: 2 | NH2: 2 | NH2: 2 |
| $B$ factors (Å$^2$) | | | | |
| Protein | 50.12 | 103.15 | 100.16 | 100.40 |
| Nucleotide | 52.14 | 116.08 | 113.98 | 115.12 |
| Ligand | 0 | 0 | 0 | 0 |
| R.m.s. deviations | | | | |
| Bond lengths (Å) | 0.003 (0) | 0.002(1) | 0.002(1) | 0.002(1) |
| Bond angles (°) | 0.525 (5) | 0.539 (6) | 0.508 (5) | 0.510(7) |
| Validation | | | | |
| MolProbity score | 1.5 | 1.38 | 1.4 | 1.4 |
| Clashscore | 4.01 | 3.08 | 3.24 | 3.26 |
| Poor rotamers (%) | 0.04 | 0.07 | 0.19 | 0.11 |
| Ramachandran plot | | | | |
| Favored (%) | 95.52 | 95.29 | 95.16 | 95.19 |
| Allowed (%) | 4.42 | 4.01 | 4.75 | 4.71 |
| Disallowed (%) | 0.06 | 0.09 | 0.06 | 0.06 |

biotin-SG-Api88(Y7B) and Api88[26], suggesting similar behaviors. Despite significant efforts, we failed to detect cross-linked complexes consisting of Api88 and ribosomal proteins by MS analysis directly. Therefore, proteins cross-linked to biotin-SG-Api88(Y7B) were enriched on streptavidin and either digested with trypsin and analyzed by mass spectrometry or separated by SDS-PAGE and detected by Western blotting with an anti-biotin monoclonal antibody (mAb, Supplementary Fig. 3). The first approach did not reliably identify proteins in the tryptic digests that were present at higher levels than in the control sample. However, the anti-biotin mAb stained bands with apparent molecular weights of ~20 kDa, ~15 kDa, ~12 kDa, and ~10 kDa at much higher intensities than in the non-cross-linked control. The corresponding bands were excised from the gel, incubated with trypsin, and analyzed by mass spectrometry, which identified the ribosomal proteins uL10, uL23, and uL29 of the large subunit as likely contact sites for biotin-SG-Api88(Y7B) (Supplementary Fig. 3, Supplementary Fig. 4 and Supplementary Fig. 5). However, the peptides from uL10 were detected with low signal intensities. In contrast to this, a previous study of Api137 identified interactions with uL22 and uL4 using a genetic screen[11]. This suggests either a different or an additional binding site for Api88, although it cannot be excluded that the 4-benzoyl-L-phenylalanine residue is not in close proximity to uL22 and uL4.

### Cryo-EM reveals different ribosome binding properties of Api137 and Api88

To further interrogate the molecular mechanism of Api88, we incubated purified 50S subunits with Api137 or Api88 and subjected the samples to cryo-EM analysis. Cryo-EM reconstructions for 50S•Api137 and 50S•Api88 complexes were refined to global resolutions of 2.64 Å and 2.44 Å, respectively (Table 1 and Supplementary Fig. 6). Closer inspection of the 50S•Api137 cryo-EM map revealed a highly defined density for Api137 in the PET (Fig. 2a). An atomic model of Api137 taken from PDB-5O2R[11] was easily fitted to this density with slight adjustments, confirming that it represents the Api137 peptide (Fig. 2b & c and Supplementary Fig. 7a). The N-terminal and central regions were located within the PET and extended to the constriction site formed by the ribosomal proteins uL4 and uL22, while the C-terminal region was located towards the PTC. To our surprise, Api88 showed a less defined density within the PET despite higher binding affinity and higher global resolution of the cryo-EM map (Fig. 2d). Superposition of the Api137 model showed that the peptide is positioned differently within the tunnel. Most notably, the local density of Api88 was more defined in the C-terminal region and branching of the local density in the central segment, close to the previously modeled Api137 residues Arg10-Arg12, and indicated the presence of two conformations, one forming an upward ('up') arc and the other forming a downward ('down') arc (Fig. 2d). Tentative modeling based on residues unambiguously assigned to the density resulted in at least three geometrically possible conformations, referred to as conformations I-III (Fig. 2e, Supplementary Fig. 7b). In addition, we noticed that the density for Api88 was located closer to the PTC than for Api137 (Fig. 2e, Supplementary Fig. 7b). The model representing conformation III was shifted 3.2 Å toward the PTC. Focused 3D classifications using different PET masks were performed to test whether the different apparent conformations could be separated into distinct subclasses (see Methods, Supplementary, Fig. 8). While most attempts converged on a single class, one attempt resulted in a more pronounced Api88 density. However, this subclass still indicated features of all modeled conformations in a subsequent refinement.

The Arg17 side chain density of Api88 appeared close to its position in the 50S•Api137 complex but extended further into a small cavity near the PTC. Modeling of the remaining peptide chain revealed the possibility of two complementary conformations adopting a down (conformation I) and an up conformation (conformation II). The shifted Arg17 side chain in conformation I resulted in a twist of the remaining chain, leaning closer to the lower PET compared to Api137 (Supplementary Fig. 9a), which was most pronounced for residues Pro11-Pro16. Conformation II similarly showed a bent C-terminus, but residues in the central region were in a more compact conformation, leaning toward the rRNA residues in the upper PET. However, density protrusion into the PTC could only be explained by a shift of the entire peptide, leading to conformation III. Rotation of Api88 conformation I by 180° and rigid-body fitting of residues Pro13-Leu18 resulted in good

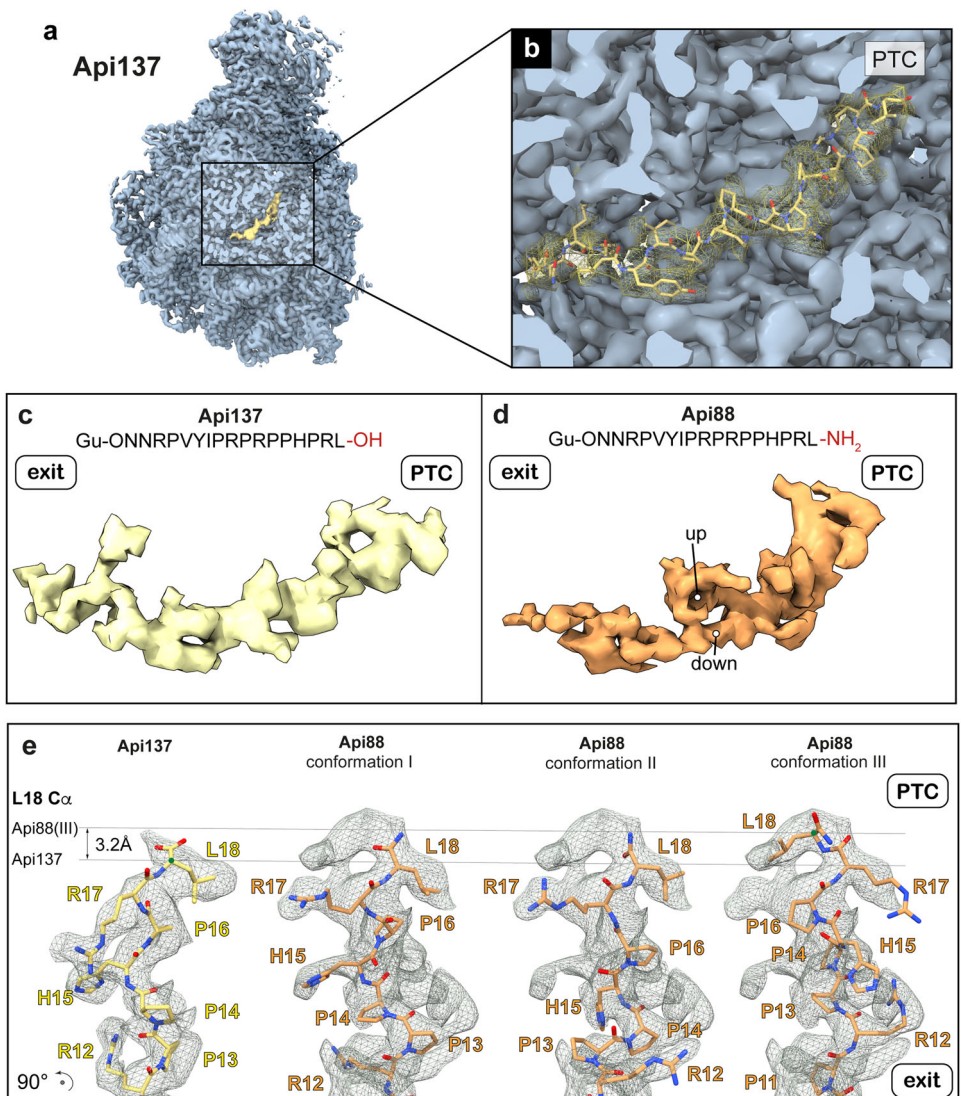

**Fig. 2 | Cryo-EM maps of 50S•Api137 and 50S•Api88 complexes. a** Cryo-EM map of Api137 (yellow) bound to the 50S subunit (blue). **b** Close-up of the Api137 binding site with local density map of Api137 (yellow mesh) and atomic model. **c** Sequence of Api137 and local density map (yellow). **d** Sequence of Api88 and local density map (orange). **e** C-terminal regions of cryo-EM maps (gray meshes) and atomic models of Api137 and Api88 (conformations I-III). The arrow indicates the 3.2 Å shift of the Api88 L18 α-C-atom in conformation III of Api88. The view is rotated by 90° relative to **c** and **d**.

agreement with the local density. This modeled conformation required the 'down' orientation.

Comparison of Api137 and the three Api88 conformations showed that the positioning of the entire peptide remains similar, but that Api88 occupies a larger available conformational space (Supplementary Fig. 9a). Especially, the central residues in the wider PET showed a greater variation of up to 6.4 Å $C_\alpha$-distances, compared to only 1.5-2 Å at the PTC. A similar tendency is observed when compared with other PrAMPs (Supplementary Fig. 9b). Onc112[9,10], bactenecin-7 (Bac7(1-16))[27,28], pyrrhocoricin (Pyr)[27,28], metalnikowin I (Met)[27,28] and drosocin (Dro)[29] explore a broad area with distances of up to 9.3 Å between α-C-atom in the region of the central Api137 and Api88 segments, while the area close to the PTC shows little variation. This suggests that the narrow region near the PTC is the key determinant and regulator of all binding modes.

## Conformational ensemble of Api88 obtained from MD simulations

To test if Api88 can indeed adopt metastable conformational states, we performed extensive all-atom MD simulations of Api88 in the PET

initiated from conformations I-III. For each of these conformations, we performed 5 simulations of 2 µs length each. Projection of the obtained trajectories onto the two dominant modes of motion shows that Api88 explores conformations close to its respective initial conformation in the simulations (Fig. 3a). The ensemble of conformations around conformation III remains far away from the other ensembles, indicating that a transition between these ensembles would take substantially longer than the µs time scale of the simulations. This transition corresponds to a shift along the tunnel axis (Fig. 2e and Fig. 3a), which is expected to have a large free-energy barrier, since many interactions between the peptide and the PET have to be broken and newly formed.

The observation that Api88 can adopt metastable states raises the question of how much they contribute to the overall conformational ensemble. To address this question, we computed a density map for each conformation (I, II, and III) from the combined trajectories initiated from that conformation. The resulting three density maps were then averaged after assigning weights $w_1$, $w_2$, and $w_3$ and the correlation coefficient with the cryo-EM map was calculated (Fig. 3b) (Supplementary Fig. 10). The weights that gave rise to the highest correlation coefficient (marked with an 'x' in the Figure) showed that all

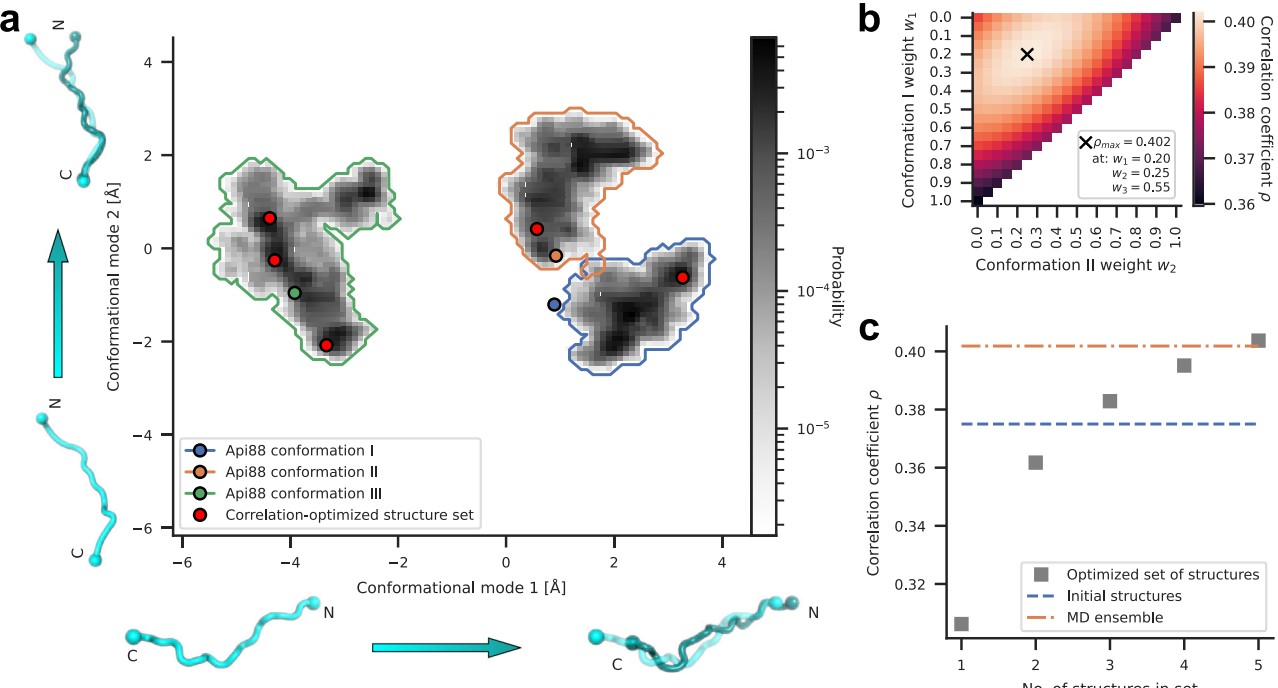

**Fig. 3 | Conformational dynamics of Api88 in the exit tunnel. a** Trajectories of Api88 obtained from MD simulations were projected onto the two dominant conformational modes of Api88 relative to the PET. The probability of observing conformations is shown in greyscale. The initial conformations I, II, and III are indicated by blue, orange, and green circles, respectively. The conformational space explored by the combined simulations from each initial conformation is outlined by colored lines. The cyan backbone pictograms along the two axes visualize the respective conformational changes (arrows). Red circles correspond to the 5 structures that optimize the correlation with the cryo-EM map (compare panel c). **b** Correlation coefficients ρ between the cryo-EM map and densities calculated from different MD ensembles are shown color coded. The ensembles were generated by assigning weights, $w_1$, $w_2$, and $w_3$, to the trajectories initiated from conformations I, II, and III, respectively. The weights corresponding to the maximum ρ are indicated by an 'x'. **c** For different numbers of structures, the maximum ρ achieved by optimally selecting structures from the simulations is shown. The ρ values obtained from the initial structures and the full MD ensemble are shown as blue and orange dashed lines, respectively.

three conformations markedly contributed to the cryo-EM density, with conformation III having the highest weight (0.55). The correlation coefficient calculated for the optimally weighted MD ensemble (0.402) is higher than that of the optimally weighted initial conformations (0.375). This increase after including the dynamics observed in the MD simulations supports the notion that Api88 is indeed dynamic and adopts an unusually large ensemble of conformations in the PET. Next, we addressed the question of how many structures are sufficient to describe the ensemble. To this end, we selected different numbers of structures from the MD ensemble to obtain the highest correlation coefficient (Fig. 3c). With a set of five structures, the correlation coefficient reached that of the MD ensemble. Notably, these structures are distributed across all sub-ensembles I, II, and III (Fig. 3a), further supporting the notion that the cryo-EM map represents an ensemble of metastable states.

**Altered electrostatics affect the interaction sites of Api88 within the 50S subunit PET**

Api137 and the putative conformations of Api88 based on cryo-EM maps exhibited overlapping and distinct contact sites within the PET (Supplementary Fig. 11a, b). Several residues of the 50S showed alternative conformations in 50S•Api137 compared to 50S•Api88, further indicating that the interactions between the peptides and the 50S are partly different (Arg92 & Arg95 of uL22, U2506 & A2062 of the 23S rRNA). Apparently, C-terminal and central interactions of Api88 rely on 23S rRNA contacts, mostly via stacking of solvent-facing nucleobases (U2504 in conformation I and II, G2505 in conformation I, U2586 in conformation III, and A2062 in conformation II and III). While A751 is the most N-terminal stacking interaction, present in all modeled Api88 conformations, other N-terminal interactions are mostly mediated by

arginine side chains of uL4 and uL22 extending toward the peptide. At the very N-terminus, several contacts are observed reaching the phosphate backbones of U1258-A1260.

The C-terminally amidated Api88 showed vastly different conformations compared to Api137, which might be largely due to the loss of the negative charge of the C-terminus leading to different electrostatic interactions within the PET (Fig. 4). Amidation abolishes the only negatively charged group increasing the overall net charge of the peptide from +5 (Api137) to +6 (Api88). The negatively charged carboxylate close to the PTC could repel Api137 from further protrusion (Fig. 4a). In contrast, all modeled conformations of Api88 were shifted further towards the PTC (Fig. 4b–d). As a consequence of these small shifts at the narrow PTC site, the remaining chain adapted and used the wider conformational space in the central region. In addition, it appears that positively charged arginine residues in the central region probe different negative rRNA sites of the tunnel. In conclusion, while all residues of Api137 reported to interact with the 50S subunit's PET are still present in Api88, C-terminal amidation appears to change the general binding mode, suggesting that the C-terminal electrostatics are important for positioning of the peptide.

Comparison of the C-terminal regions suggested a structural rationale (see discussion) for the RF-independent binding mode of Api88 (Supplementary Fig. 12). The Api137 structures within vacant 50S and 70S in the presence of RF1[11] are geometrically highly compatible (Supplementary Fig. 12a, b, e). In the absence of RFs, Arg17 formed a hydrogen bond to the backbone of G2505. In the presence of RF1, the C-terminus shifted slightly into the tunnel. In conformations I and II of Api88, Arg17 is involved in a stacking interaction with U2504, leaving the side chain in the correct position (Supplementary Fig. 12c, f).

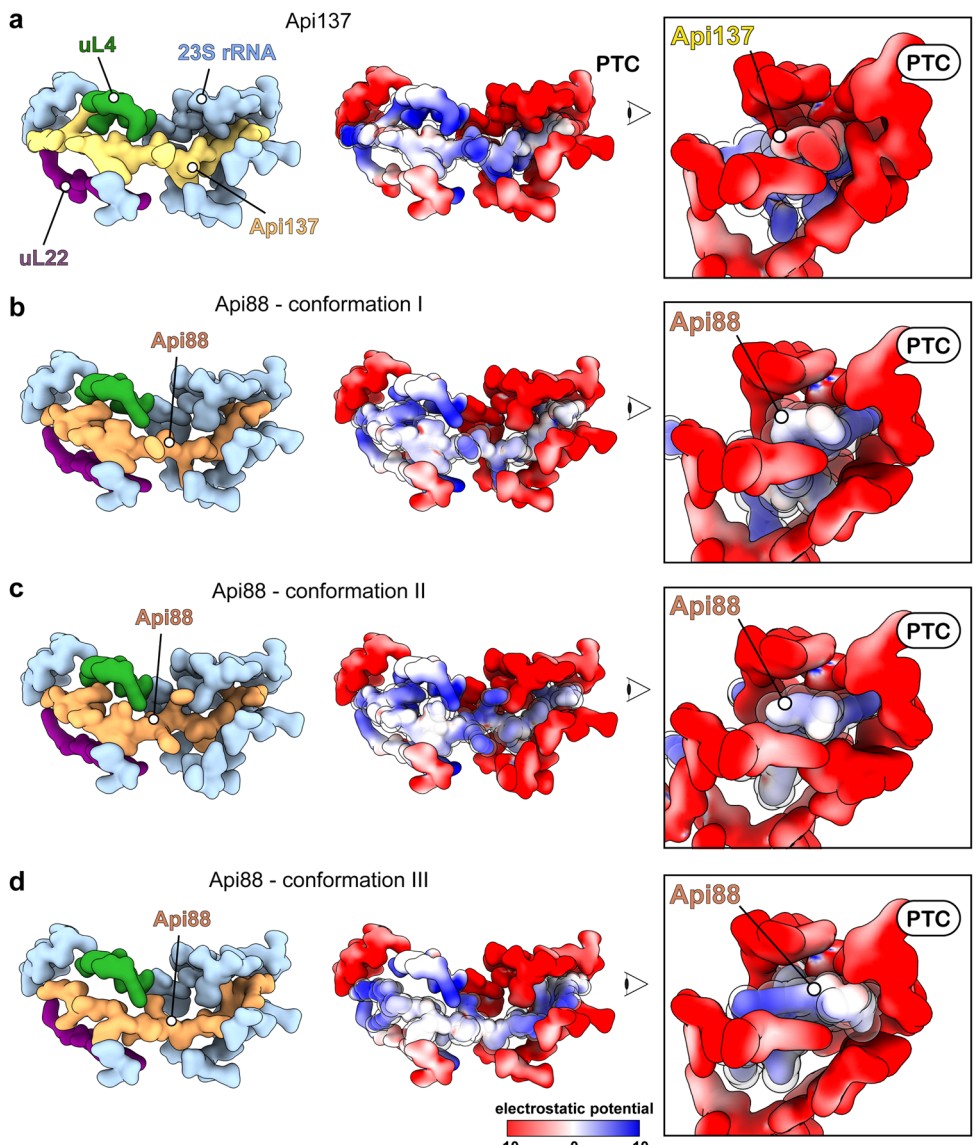

**Fig. 4 | Electrostatic interactions of apidaecins within the PET.** Surface models with labels (left), electrostatic surface potential (middle), and view from the PTC into the PET (right) for **a** 50S•Api137, **b** 50S•Api88 conformation I, **c** 50S•Api88 conformation II, and **d** 50S•Api88 conformation III. Coulombic electrostatic potentials are shown in red (electronegative; −10 kcal/mol·*e*), white (neutral; 0 kcal/mol·*e*) and blue (electropositive; 10 kcal/mol·*e*). Surface models are calculated at 4 Å resolution. For clarity, Api137 and Api88 are highlighted using a transparent sphere model in electrostatic representations.

In contrast, conformation III of Api88 is rotated and shifted compared to Api137 (Supplementary Fig. 12d, g). This conformation would clash with the RF and has Arg17 positioned on the opposite side. Besides the potential clashes and missing interaction sites, it could be hypothesized that the dynamic binding of Api88 impedes proper trapping of the RF prior to its dissociation.

## A second binding site at the end of the PET

Careful inspection of the 50S•Api137 and 50S•Api88 cryo-EM maps revealed an additional density corresponding to another apidaecin molecule in an unexpected location near the exit of the PET (Fig. 5a, b). The peptide extended to the end of the PET and attached horizontally to the solvent-exposed nucleobases A1321, A507, A508, A63, A92, and A93 (Fig. 5c). The density was defined in the 50S•Api137 complex and allowed modeling of the peptide chain from Asn2 to Leu18 (Fig. 5d). The N-terminal region was located at the exit pore close to the ribosomal proteins uL29 and uL23 (Supplementary Fig. 13a), reasoning the observed cross-links of Api88. The C-terminus protruded further into the tunnel where the C-terminal region showed multiple interactions.

The binding of Api137 was mediated by multiple hydrogen bonds and stacking interactions (Fig. 5 e–g). Asn3 of Api137 binds via a hydrogen bond to an oxygen and a hydroxyl group of ribose U62, while the side chain of Val6 is positioned on top of A63. Tyr7 is wedged between A91 and A92, potentially stabilized by stacking (Fig. 5g). Arg10 binds to A507 through a stacking interaction in this highly solvent-exposed region. In the C-terminal region, Pro13 and Pro16 are potentially stacked on A508 and A1321, respectively. Interestingly, the His15 side chain extends to Pro11 and stabilizes the structure intramolecularly. The C-terminal carboxylate forms a hydrogen bond with the ribose oxygen of A1319. In addition, electrostatic interactions of positively charged residues in the C-terminal segments mediate the binding to the 23S rRNA (Fig. 5h–j).

The 50S•Api88 complex was less defined, but fitting the modeled peptide into the local density indicated that the binding modes are very similar (Supplementary Fig. 13b–j). The different C-termini of Api88 and Api137 therefore did not appear to alter the overall binding mode within this binding site, although the C-terminal hydrogen bond is not formed between Api88 and the rRNA.

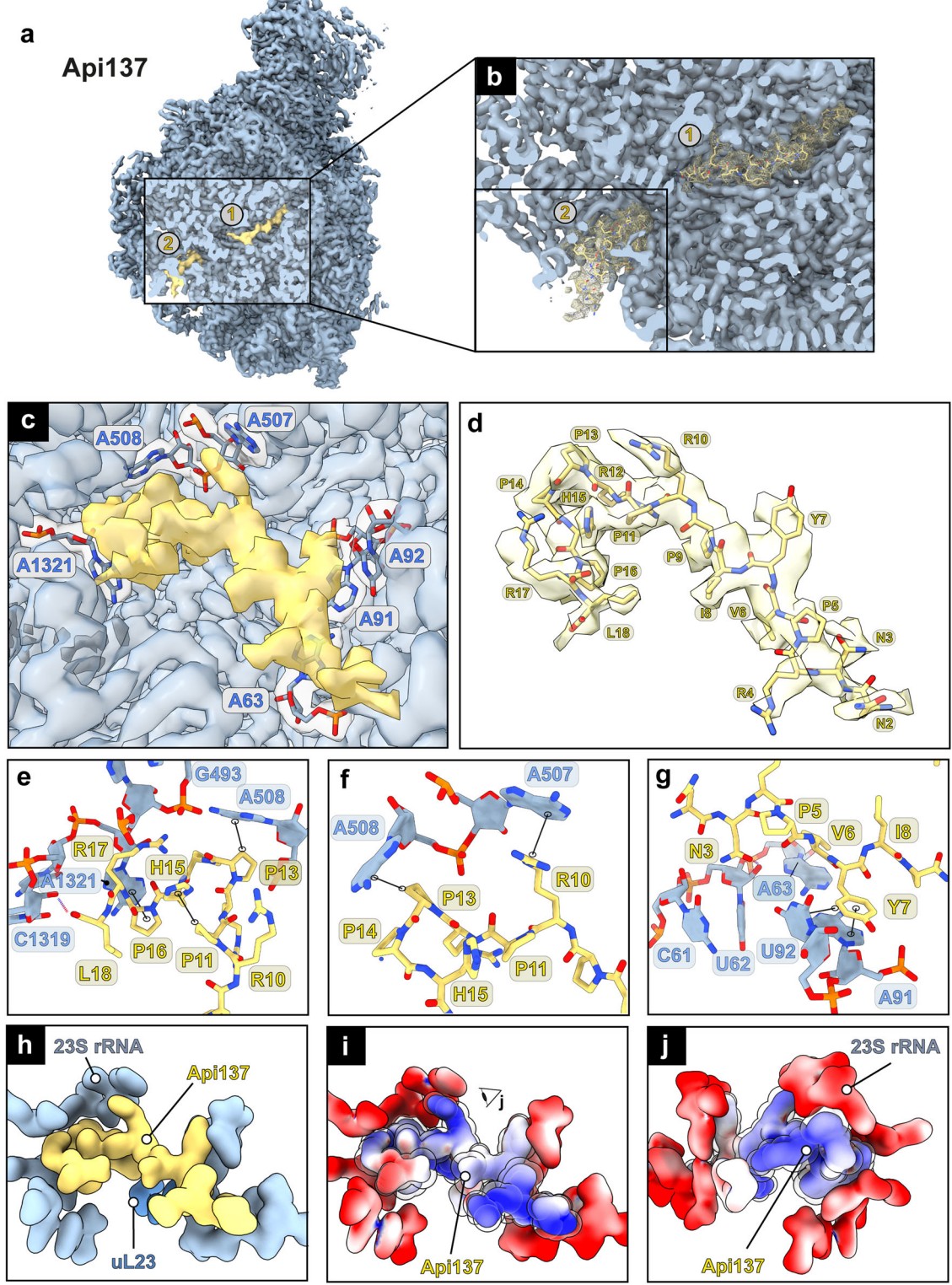

**Fig. 5 | Additional binding site of Api137 at the PET exit. a** Cryo-EM map of the 50S•Api137 complex with two Api137 molecules bound **b** close-up with Api137 densities (yellow mesh) and atomic models. **c** Assigned Api137 density in yellow illustrating rRNA interaction sites. **d** Modeled Api137 residues in the local density map. **e–g** More detailed interactions. Hydrogen bonds are shown as dashed lines, stacking interactions as connected black circles. **h** Calculated surface models of Api137 and surrounding rRNA at 4 Å resolution. **i** Electrostatic surface potential of Api137 and surrounding rRNA sites. Coulombic electrostatic potentials are shown in red (electronegative; −10 kcal/mol·*e*), white (neutral; 0 kcal/(mol·*e*) and blue (electropositive; 10 kcal/mol·*e*). **j** Rotated view of **i**. For clarity, Api137 is highlighted using a transparent sphere model in electrostatic representations.

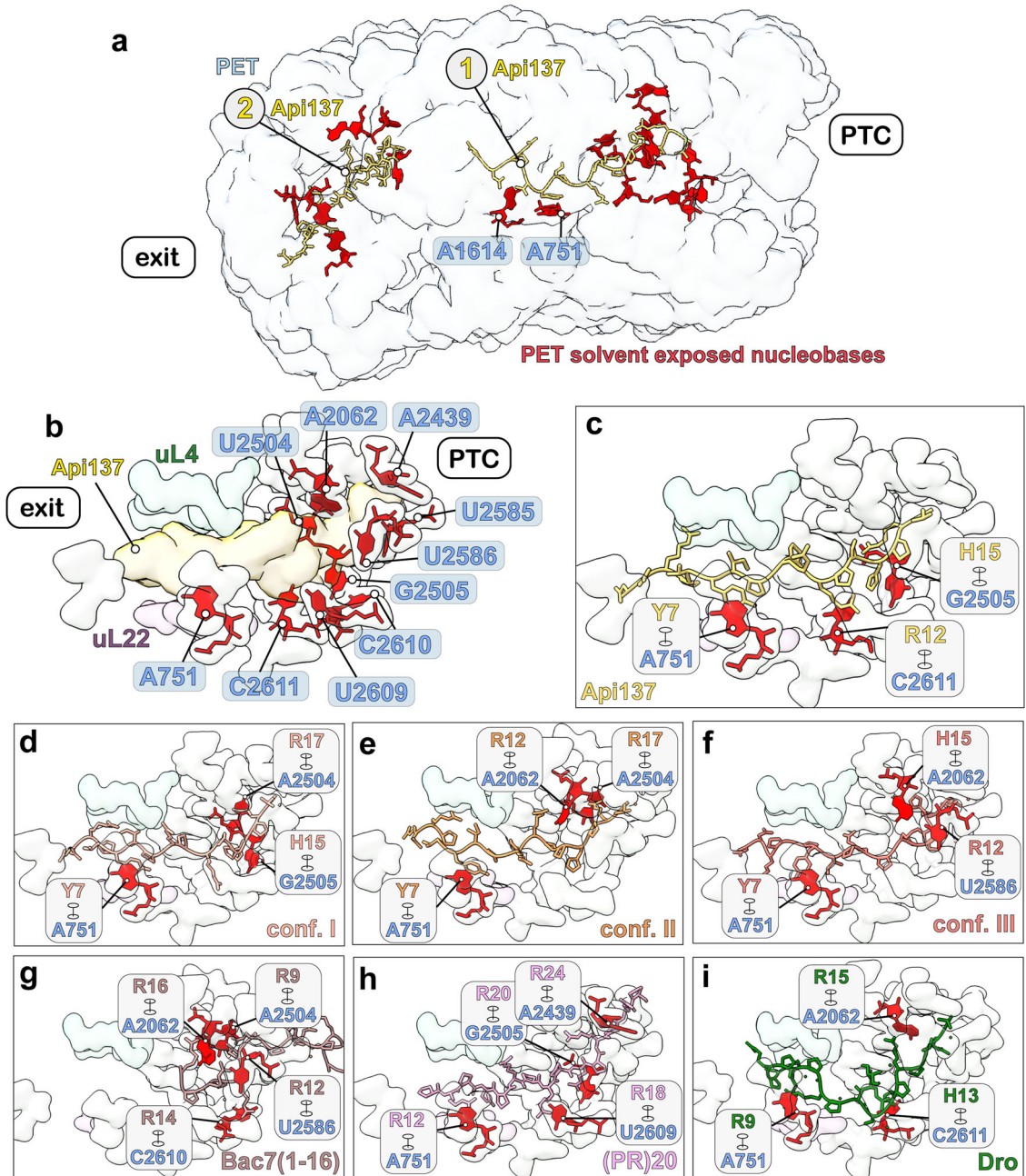

**Fig. 6 | Binding sites of Api137, Api88, and other PrAMPs co-localize with solvent-exposed nucleobases. a** All solvent-exposed nucleobases (red) within the PET and Api137 in binding sites 1 and 2 are shown. Api137 (yellow) co-localizes with these 23S rRNA residues. The PET is shown as the simulated map at 10 Å resolution. **b** Solvent-exposed bases at the PTC site are shown in red. Simulated surface maps of Api137 (yellow) and its surrounding residues are simulated 4 Å and are shown as transparent volumes. **c–i** Stacking interactions to specific rRNA bases by residues of **c** Api137 (yellow); **d** Api88 conf. I (salmon); **e** Api88 conf. II (orange); **f** Api88 conf. III (light red); **g** Bac7(1-16) (light brown), PDB: 5F8K[27,28]; **h** (PR)20 dipeptide (pink), PDB: 7TOS [30]; **i** Dro (green), PDB: 8ANA[29],. Views in **c-i** are slightly adapted to increase visibility.

A striking observation is that both binding sites of Api137 and Api88 exhibit solvent-exposed nucleobases of the 23S rRNA (Fig. 6). While the central region of the tunnel is formed by backbone residues of the rRNA or long extensions of the ribosomal proteins uL4 and uL22, the PET close to the PTC and near the exit shows several of these exposed, in some cases flexible, bases (Fig. 6a). Both Api88 and Api137 co-localize with these sites and appear to use these bases as binding hubs, by forming π-interactions via their Arg, His, and Tyr side chains (Fig. 6b–f). This seems to be a common strategy for PrAMPs to bind to the region close to the PTC, as all peptides use a different combination of stacking interactions with rRNA bases (Fig. 6g–i)[9,10,27–30].

## A third binding site of Api88 within domain III of the 23S rRNA

Upon further investigation, we identified a third binding site for Api88, but not for Api137, located deep within domain III of the 23S rRNA with parts of the Api88 density highly resolved (Fig. 7a–c). Although the density appeared weaker compared to binding sites 1 and 2, focused classification did not result in a separation of peptide-bound particles. The peptide was positioned with its N-terminus toward the PET and the C-terminus toward the center of the subunit. The well-resolved central region of the peptide is located above helix H51 and below the phosphate backbone of helix H49 of the 23S rRNA (Fig. 7d–f). Arg10 and Arg12 form hydrogen bonds with the phosphate backbone of A1346 and

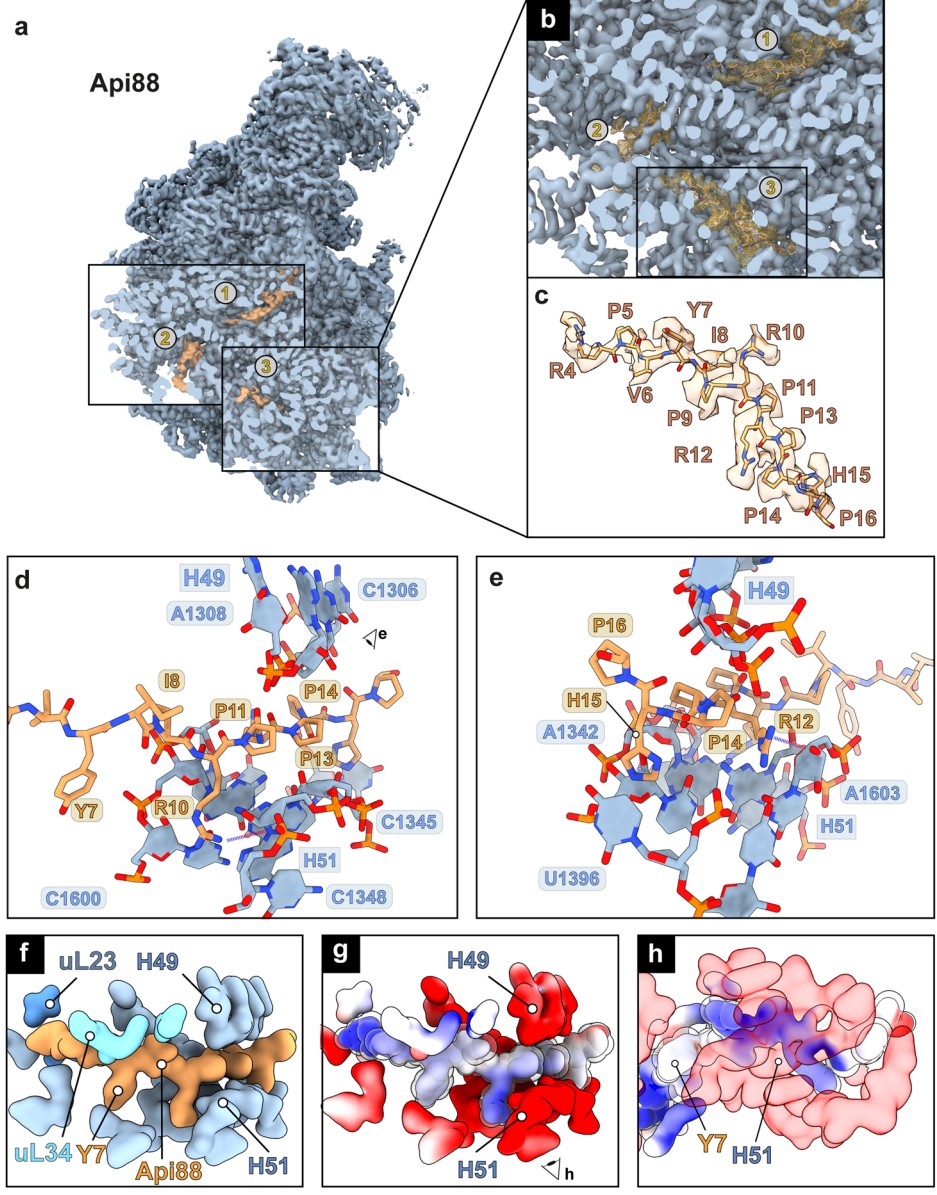

**Fig. 7 | Third binding site of Api88 within domain III of the large ribosomal subunit. a** Cryo-EM density map of 50S•Api88 and the location of the three identified Api88 binding sites (orange) on the 50S subunit (blue). **b** Cross section along the Api88 tunnel, tunnel end and third binding site. **c** Cryo-EM density map of Api88 (orange) in binding site 3 and corresponding atomic model. **d**−**e** Molecular interactions of Api88 with surrounding rRNA residues. Hydrogen bonds are shown as dashed lines. **f** Calculated surface model of Api88 in its third binding site, sandwiched by rRNA helices H49 and H51 at 4 Å resolution. **g** Electrostatic surface potentials of the respective region. Coulombic electrostatic potentials are shown in red (electronegative; −10 kcal/mol·*e*), white (neutral; 0 kcal/mol·*e*) and blue (electropositive; 10 kcal/mol·*e*). **h** Rotated view of g from below H51. For clarity, Api88 is highlighted using a transparent sphere model in electrostatic representations.

A1603. The proline residues position the peptide ideally between the rRNA helices H51 and H49. His15 forms a hydrogen bond with the ribose oxygen of C1345. The weakly resolved N-terminal region is located near Arg76 of uL23 and the backbones of 23S rRNA residues C57 and U65. The Coulomb potentials suggest electrostatic interactions between the positively charged peptide and the negatively charged rRNA backbone as a main driving force of this binding event (Fig. 7g, h). Taken together, the central Pro9-Arg10-Pro11-Arg12 motif appears to be the main determinant for stable positioning of Api88 between helices H51 and H49, rather than stacking interactions.

### Proline residues confer stability

The presence of apidaecin peptides at different locations within the 50S subunit allowed us to compare the conformational stability of the peptides. Therefore, the N-terminal, central, and C-terminal parts of models of Api137 at binding site 2 and Api88 derived from its three confirmations in binding site 1 and from binding site 3 were aligned to the canonical Api137 conformation within the PET (Supplementary Fig. 14). While the overall geometry differed between the models, the peptide retained most of its conformation in the proline-rich regions. The conformational deviations were most pronounced at residues involved in putative direct interactions with the 50S subunit. The N-terminal Asn2-Asn3-Arg4 motif showed the strongest deviation and explains why this part is the least resolved, also for Api137. Hence, another reason for the well-defined densities of Api137 and Api88 in their different binding sites is the rigidity of the peptides, which is mostly determined by the proline residues.

## Discussion

Api137 and Api88 are highly similar derivatives of the naturally occurring apidaecin 1b. In Api88 the C-terminus is additionally amidated, which removes the only negative charge within the peptide, and increases its cationic character. Our reports identifying the bacterial ribosome as a major target of PrAMPs and mapping their binding site to the nascent PET[7,8] triggered structural analyses that led to the identification of the binding site and mode of action of Onc112[9,10] and shortly thereafter, Api137, which arrests protein translation by trapping RF1[11]. This mechanism was assumed to apply also to Api88 despite its neutral amidated C-terminus, since the C-terminal carboxyl group of Api137 does not interact directly with the 50S subunit or RF1 (but with the P-site tRNA). However, the present work challenged the supposed mechanism of Api88 by in vitro protein translation experiments showing that the entire inhibitory effect of Api88 and half of the inhibitory effect of Api137 is based on a RF1-independent mechanism. The different binding modes were further supported by the 10- to 20-fold higher $K_d$ values obtained for Api137 after purification of ribosomal extracts to remove RFs and other contaminating proteins, whereas the $K_d$ value of Api88 remained rather stable, similar to Onc112. In addition, high-resolution cryo-EM reconstructions of purified 50S subunits incubated with Api137 or Api88 confirmed multiple binding sites for both peptides. In this respect the reported $K_d$ values should be considered as apparent $K_d$ values, as they represent the average $K_d$ of at least two or three binding sites of Api137 and Api88, respectively, although there may be additional binding sites not visible in the structures. The binding curves indicated that cf-Api88 binds to the ribosome and that Api88 can completely displace it, as indicated by the significant decrease in fluorescence polarization, while erythromycin reduced the fluorescence polarization to a much lesser extent, suggesting at least one further binding site of Api88 not occupied by erythromycin.

Our cryo-EM and MD data suggest the following modes of action for Api137 and Api88: The positively charged apidaecins are electrostatically attracted to the overall negatively charged ribosome (Fig. 8a). The Arg side chains of the peptides perform a dual function. They contribute to the positive net charge, and they can participate in π-interactions with nucleobases. The PET exit has many nucleobases exposed on the surface and available for stacking interactions with the Arg, His15, Tyr7, and Pro residues (binding site 2). Upon release of the nascent chain and following diffusion as well as the negative net charge of the PET interior, Api137 and Api88 migrate into the tunnel that narrows like a funnel. Beyond the central constriction formed by uL4 and uL22, an array of exposed nucleobases protrudes into the interior of the upper tunnel (Figs. 5c and 6). One of these nucleobases, the conserved A751, appears to be an initial anchor point within the PET. Following the electric charge, the peptides move along the nucleobase rungs, π-interacting with the amino acid side chains. At some point the movement stops for Api137, while Api88—owing to its stronger net positive charge—continues to migrate further towards the PTC, exhibiting an ensemble of conformations (binding site 1). The different conformations may represent the crawling motions that apidaecins utilize to enter the PET. Due to its stable positioning, the C-terminal carboxyl group of Api137 is able to interact with the RFs and arrest them during termination (Fig. 8b). Api88 remains in a more dynamic metastable state, which allows the dissociation of RFs and P-site tRNA (Fig. 8c). In addition, Api88 can adopt a unique binding site (binding site 3) within domain III of the 23S rRNA (Fig. 8d). The negatively charged C-terminus of Api137, instead, likely hinders passage through the narrow cavity between negatively charged helices H51 and H49 of the 23S rRNA.

How does this proposed mechanism involving exposed nucleobases relate to what is known about other PrAMPs? Remarkably, all conformations of Api88 were found to be more similar to Onc112[9,10] and Bac7(1-16)[27,28] than to Api137, particularly in the positioning of Pro11-Arg12-Pro13, which was comparable to Pro10-Arg11-Pro12 in Onc112 (Supplementary Fig. 9c–e) or Pro13-Arg14-Pro15 in Bac7(1-16) (Supplementary Fig. 9f). This resembles the general Pro-Arg-Pro motifs characteristic of all PrAMPs. However, it is unclear whether these peptides use a similar entry mechanism as they extend further into the PTC and also inhibit translation during elongation. Interestingly, Dro1, which also arrests RFs, is the only peptide that extends similarly to the PET. It shares a stacking interaction at the 23S rRNA residue A751 (Supplementary Fig. 9g), suggesting that this interaction is crucial for stabilizing peptides in this region[29].

In contrast to Onc112 and Bac7(1-16), Api88 and Api137 bind to the PET in the same orientation as the nascent protein chain (N-terminus toward the exit of the PET) and appear to use mechanisms reported for the self-inhibitory effects of proteins containing RxPP, repetitive PR, and polyproline motifs[30–32]. Consequently, these motifs are rarely found in proteins. They can even trigger diseases, such as the repetitive PR motifs identified in toxic dipeptide-repeat (DPR) peptides, a common genetic cause of amyotrophic lateral sclerosis and frontotemporal dementia[30]. Sequences containing at least 20 PR repeats can inhibit protein translation at sub-micromolar concentrations by binding to the PET region near the PTC in both bacterial and mammalian ribosomes. This binding region is also occupied by much shorter PrAMPs that contain characteristic PRP and PP motifs, such as the PRPRPPHPR sequence in Api137 and Api88. This region has been described as the critical pharmacophore unit for antibacterial activity of apidaecins[12,33].

Binding sites two and three are close to uL23 and uL29, which is also supported by Api88(Y7B) cross-links. These ribosomal proteins are located adjacent to the PET exit and are binding hubs for the molecular chaperone trigger factor (TF)[34] and the signal recognition particle (SRP), a universally conserved and essential cellular machinery that directs proteins destined for membrane integration to the plasma membrane[35,36]. This co-translational targeting is initiated by a hydrophobic signal sequence in the nascent protein chain recognized by SRP as it exits the PET. It is appealing to hypothesize that Api137 and Api88 occupying binding site two, and Api88, located in its unique binding site three, interfere with TF and SRP-uL23 interactions, presumably via induced structural changes in uL23. Interestingly, TF was reported as a potential interaction partner of Api88 besides ribosomal proteins and other proteins, including DnaK[17,26], based on the above-mentioned cross-linking experiments using Api88(Y7B)[26]. However, studies using fluorescence polarization assays could not confirm that Api88, Api137 or apidaecin 1b bind to recombinant TF ($K_d$ > 100 μmol/L)[37]. In the context of the new structures, this data may support the in vivo relevance of the identified second and third binding sites to disturb TF-ribosome binding, as Api88 can occupy parts of the TF-ribosome contact area represented by uL23 and uL29. This is also consistent with the RF-independent basal inhibition of sfGFP expression observed in iTT. It is tempting to hypothesize that Api137 captures RF1 at the ribosome, while Api88 captures TF or disturbs TF-ribosome interactions. Similarly, inhibition of SRP-uL23 interactions could prevent the proper localization of membrane proteins, which represent about one-third of the bacterial proteome, to the plasma membrane and induce their intracellular aggregation. Alternatively, Api88 could trap SRP at the ribosome, preventing the proper localization of membrane proteins synthesized on other ribosomes, or trap the ribosome-SPR complex at the plasma membrane. Even if this mechanism is not as efficient as the mechanism used by Api137, the higher uptake rates of Api88 may compensate for this due to higher concentrations in the cytoplasm[15,16,20].

It was surprising and exciting to explore how a small chemical modification (i.e. an amidation of the C-terminus) of the 18-residue long Api137 altered the mechanism of action of the resulting Api88 while providing comparable antibacterial activity. This may guide future designs aimed at combining the beneficial effects of amidation

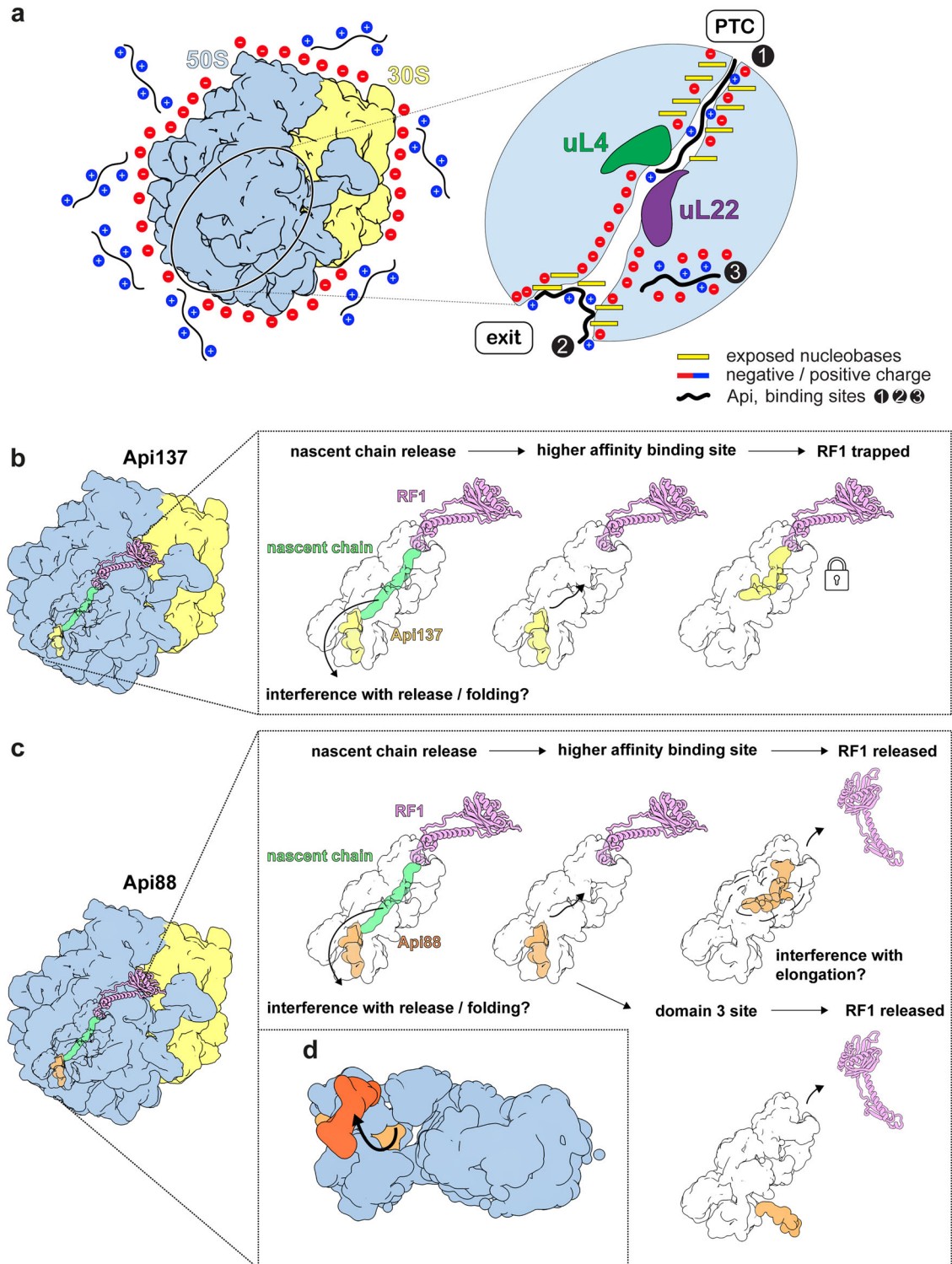

**Fig. 8 | Models for the molecular mechanisms of Api137 and Api88. a** Schematic model describing how the two peptides approach and enter the 50S subunit. The cationic peptides Api137 and Api88 can associate with the negatively charged rRNA of the 70S ribosome. The close-up shows a schematic view of the PET. The PET forms a funnel towards the PTC site and exhibits several solvent-exposed nucleobases at the PET exit and near the PTC. The three binding sites are indicated (1, 2, and 3). **b, c** Structure-based models for peptide entry into the PET and modes of action. **b** Api137: Occupation of binding site 2 at the exit pore of the PET by Api137 (or Api88) could still allow nascent polypeptide chains to pass. Api137-mediated

release factor-dependent inhibition of translation, as proposed by Florin et al.[14] Upon release of the nascent chain, Api137 interacts with RF1, which is thereby trapped. **c** Api88: Upon release of the nascent chain, Api88 adopts multiple conformations within the PET and is unable to interact with RF1. Instead, the binding sites in the PET could interfere with regular translation. Both Api137 and Api88 utilize at least two binding sites, consistent with a "sneak-up" model. **d** The putative entry region for Api88 within the PET to its third binding site is shown (light orange: Api88 in the second binding site, dark orange: Api88 in the third binding site).

in Api88 with the trapping of RF1 observed only for Api137. The implications of Api88 interacting with its third binding site should be further evaluated as it might represent a novel, unexplored, potentially bactericidal mechanism affecting the interaction of the highly conserved SRP with the ribosome.

## Methods

### Reagents

Biosolve BV (Valkenswaard, Netherlands): Acetonitrile (ULC/MS grade), formic acid (≥99%), N,N-Dimethylformamide (DMF, >99.8%), and piperidine (>99.5%); Carl Roth GmbH & Co. (Karlsruhe, Germany): Dichloromethane (DCM, 99%), trifluoroacetic acid (TFA, ≥99.9%, peptide synthesis grade), Lysozyme, sodium dodecyl sulfate (SDS), and glycerin (99.9%); Honeywell Fluka™ (Seelze, Germany): Ammonium bicarbonate (99.5%), 1,3-disopropyl-carbodiimide (DIC, ≥98.0%), N,N'-diisopropylethylamine (DIPEA, 98%), 1,2-ethandithiole (≥98%), $MgCl_2$ (99%), and thioanisole (≥98%); Sigma Aldrich (Steinheim, Germany): Ammonium chloride ($NH_4Cl$, 99.5%),4-benzoyl-L-phenylalanin (BPA, 98%), 5(6)-carboxyfluorescein (Cf, for fluorescence), casein, m-cresol (98%), 1-hydroxybenzotriazole hydrate (HOBt, ≥97.0%), hydrochloric acid (HCl), 2-mercaptoethanol (2-ME, 99%), N-methylmorpholine (NMM, 99.5%), potassium dihydrogen phosphate ($KH_2PO_4$, 99.5%), potassium hydroxide (KOH, 90%), sodium chloride (NaCl, 99.5%), sodium hydrogen phosphate ($Na_2HPO_4$, 99%), triisopropylsilane (TIS, >98.0%), and TFA (≥99%, HPLC grade); VWR International GmbH (Dresden, Germany): Acetonitrile (≥99.9%), diethyl ether (99.9%), and formic acid (FA, -98%, LC-MS grade); AppliChem GmbH (Darmstadt, Germany): HEPES (99.5%) and tris(hydroxymethyl)aminomethane (TRIS); SERVA (Heidelberg, Germany): Acrylamide/bisacrylamide solution (37.5:1), ammonium persulfate (APS), Coomassie Brilliant Blue G250, tetramethylethylenediamine (TEMED, 99.9%), trypsin, and Tween20; Iris Biotech GmbH (Marktredwitz, Germany): D-Biotin and 2-(1H-benzotriazol-1-yl)1,1,3,3-tetramethyluronium hexafluorophosphate (HBTU); Riedel-de Haën (Seelze, Germany): Bromphenol blue.

All 9-fluorenylmethyloxycarbonyl- (Fmoc-) protected amino acid derivatives used for peptide synthesis were obtained from Orpegen Pharma GmbH (Heidelberg, Germany) or Iris Biotech. Water (resistance ≥18 mΩ, total organic content <1 ppb) was purified in-house using a PureLab Ultra Analytic system (ELGA Lab Water, Celle, Germany).

### Peptide synthesis

The peptides were synthesized on solid phase using a multiple synthesizer (SYRO2000, MultiSynTech GmbH, Witten, Germany), Fmoc-chemistry, in situ activation with DIC in the presence of HOBt, and Rink amide or Wang resins to obtain C-terminal peptide amides or acids, respectively. The N-termini of both peptides were guanidated with HBTU in the presence of NMM. Ornithine was incorporated as 4-methyltrityl (Mtt) -protected derivative (Fmoc-Orn(Mtt)-OH) and the Mtt-group was selectively cleaved with 2% (v/v) TFA and 2.5% (v/v) TIS in DCM. When necessary, 5(6)carboxyfluorescein (cf) was coupled with HBTU in the presence of DIPEA to the N-terminus of Onc112 or the δ-amino group of Orn1 in Api137 and Api88. For the cross-linking experiments Tyr7 was substituted with 4-benzoyl-L-phenylalanin (Bpa, B) and a Biotin-Ser-Gly-linker coupled to the δ-amino group of Orn1 to obtain biotin-SG-Api88(Y7B) and biotin-SG-Api137(Y7B). Isotope-labeled peptides were synthesized by manual coupling of Fmoc-L-Pro($^{13}C_5$, $^{15}N$)-OH in positions 5 and 9 using DIC in the presence of HOBt. Peptides were cleaved with TFA containing 12.5% (v/v) scavenger mixture (ethanedithiol, m-cresol, thioanisole, and water; 1:2:2:2 (by volume)) and precipitated with cold diethyl ether. Peptides were purified on an Äkta Purifier 10 using a Jupiter $C_{18}$-column (ID 21.2 mm) with an aqueous acetonitrile gradient in the presence of 0.1% TFA as an ion pair reagent. Purity was determined by RP-HPLC using a Jupiter $C_{18}$-column (ID 4.6 mm or 2 mm). Molecular weights were confirmed by

matrix-assisted laser desorption/ionization time-of-flight mass spectrometry (MALDI-TOF-MS; 5800 Proteomic Analyzer; AB Sciex, Darmstadt, Germany) or by ESI-MS (Esquire HCT; Bruker Daltonics, Bremen, Germany or Q Exactive plus™, Thermo Scientific, Henningsdorf, Germany), analytic control can be seen in Supplementary Fig. 15 and 16.

### In vitro transcription/translation assay and toeprinting analysis

sfGFP was expressed using the NEB PureExpress Delta RF123 Kit (New England Biolabs). The sfGFP DNA template encoding for an UAG stop codon was amplified from the pY71sfGFP plasmid by PCR using sfGFP primers (sfGFP fwd: TAATACGACTCACTATAGGG and sfGFP rev: CATGAAGCTTATTTTTCGAACTGCGGAT). The release factor RF1 included in the kit was 50-fold diluted (v/v) in freshly prepared 1x Pure System Buffer (PSB, 9 mmol/L magnesium acetate, 5 mmol/L $K_3PO_4$ (pH 7.3), 95 mmol/L potassium glutamate, 5 mmol/L $NH_4Cl$, 0.5 mmol/L $CaCl_2$, 1 mmol/L spermidine, 8 mmol/L putrescine, 1 mmol/L DTT). sfGFP was expressed in a freshly prepared mixture of kit solution A (2 μL), kit solution B (1.5 μL), 50-fold diluted RF1 (0.5 μL), sfGFP template (0.1325 pmol, 35 ng) or water as control (0.25 μL), PSB (0.25 μL), and a PrAMP (50, 5, 0.05 or 0 μmol/L) dissolved in water (0.5 μL). The reaction mixture was transferred to a black 384-well plate (Greiner Bio-One GmbH, Frickenhausen, Germany), covered with a lid, and incubated in a microplate reader (Gemini EM, Molecular Devices) at 37 °C for two hours. Fluorescence was recorded every 10 min ($\lambda_{exc}$ = 485 nm, $\lambda_{em}$ = 535 nm).

Toeprinting experiments using the *yrbA-fs15* DNA template with a TAG stop codon were performed in the NEB classic PURExpress system (New England Biolabs)[21] as described above. The template was prepared by PCR[21] by combining the overlapping primers T7, T7-IR-AUG, IR-yrbA-fs15-RF1 (for TAG), post-NV1, and NV1, rendering the sequence:

TAATACGACTCACTATAGGGCTTAAGTATAAGGAGGAAAACAT<u>ATGATATACCCCTGCGGAGTGGGCGCGCGATCGCAAACTGAACGGCTT(**TAG**)</u>GCCGACCTCGACAGTTGGATTCACGTGCTGAATCCTGATGCGATGTCGAGTTAATAAGCAAAATTCATTATAACC (ORF sequence is underlined). The peptides Api137 and Api88 were added from stock solutions in water to a final concentration of 50 μM.

### Ribosome preparation

*E. coli* BW25113 were grown in LB medium and harvested by centrifugation (5000× *g*, 15 min, 4 °C) at an $OD_{600}$ of 3-4. Cell pellets were washed in ribosome preparation buffer (20 mmol/L HEPES-KOH, 6 mmol/L $MgCl_2$, 30 mmol/L $NH_4Cl$, pH 7.6, 4 °C), centrifuged (4100× *g*, 30 min, 4 °C), and suspended in ribosome preparation buffer (0.5 g/L) containing freshly added ß-mercaptoethanol (4 mmol/L). Lysozyme was added to the resuspended cells and incubated for 30 min on ice. Cells were disrupted in six cycles using a FastPrep-24™ 5 G instrument (MP Biomedicals, Eschwege, Germany) with the BigPrep 50 mL setting (40 s, 4.0 m/s) and one-minute incubation on ice between cycles. After centrifugation (1620× *g*, 5 min, 4 °C), the supernatant was incubated with DNase I (5 U/mL) for 1 h on ice and centrifuged at 16,000× *g* (30 min, 4 °C) and twice at 32,000× *g* (60 min, 4 °C) to remove cell debris. Crude ribosomes were sedimented at 165,000× *g* (17 h, 4 °C). The pellet was washed with ribosome preparation buffer (1 mL), suspended in ribosome preparation buffer (1 mL), and this 70S ribosome extract was stored at -80 °C.

Ribosomes were purified by anion exchange chromatography using an Äkta Pure System (GE Healthcare, Freiburg, Germany) and a CIM®QA column (8 mL, 2 μm channels; Sartorius Stedim Biotech GmbH, Göttingen, Germany). The ribosome preparation buffer was set as eluent A while eluent B had an elevated $NH_4Cl$ (1 mol/L) concentration (20 mmol/L HEPES-KOH, 6 mmol/L $MgCl_2$, 1 mol/L $NH_4Cl$, pH 7.6, 4 °C). The ribosome pellet was diluted in 40 % eluent B, applied to the column at a flow rate of 2 mL/min and eluted at a flow rate of 8 mL/min using a linear gradient from 40% to 100 % eluent B in 3 min.

The ribosome fractions were combined, concentrated, and rebuffered in ribosome preparation buffer with freshly added 2-mercaptoethanol (4 mmol/L) using Amicon®Ultra-15 centrifugal filters (Merck Millipore, Darmstadt, Germany). The ribosome concentration was estimated by recording the absorbance at 260 nm (considering that 15 AU corresponds to a ribosome concentration of 1 g/L). The molecular weight of the 70S *E. coli* ribosome was assumed to be 2.3 MDa. The purity was checked by SDS-PAGE (Supplementary Fig. S1).

## Large-scale preparation of 50S and 30S ribosomal subunits

Ribosomes were isolated from the *E. coli* strain MRE600, which does not produce RNase I due to a missense mutation in the *rna* gene[38]. Several hundred grams of log-phase cells were commercially produced and purchased. Cells were washed in buffer A (20 mmol/L HEPES (pH 7.6), 6 mmol/L MgAc$_2$, 30 mmol/L AcOK, 4 mmol/L ß-mercaptoethanol) and lysed in an M-110L microfluidizer (Microfluidics) at 17,000 psi and 4 °C. Cell debris was removed by low-speed centrifugation (15,000× *g*, 10 min, 4 °C) to obtain S30 lysate. Crude ribosomes, mostly containing 70S ribosomes, were pelleted by ultracentrifugation (40,000× *g*, 17 h, 4 °C) and resuspended in buffer A. The 70S ribosomes were separated from 30S and 50S subunits by zonal ultracentrifugation (Beckman Ti-15; 6–40% sucrose gradient in buffer A, 44,000× *g*, 16 h, 4 °C). The 70S-containing fractions were collected, and the ribosomes were sedimented by ultracentrifugation (40,000× *g*, 17 h, 4 °C). Since these 70S ribosomes still contained tRNAs and mRNA, they were resuspended in buffer B (20 mmol/L HEPES (pH 7.6), 1 mmol/L MgAc$_2$, 0.2 mol/L NH$_4$Ac, 4 mmol/L ß-mercaptoethanol), which allows for dissociation into 30S and 50S subunits, and subjected to a second zonal centrifugation (Beckman Ti-15, 44,000× *g*, 16 h, 4 °C), but this time in a 6–40% sucrose gradient prepared in buffer B. Fractions containing 30S and 50S subunits were collected, sedimented by ultracentrifugation (40,000× *g*, 17 h, 4 °C), resuspended in buffer A, aliquoted, shock-frozen in liquid nitrogen, and stored at −80 °C.

## Fluorescence polarization

Dissociation constants ($K_d$) were determined using purified 70S ribosome, 50S or 30S subunit and cf-labeled peptides in ribosome preparation buffer. Briefly, black 384-well plates (flat bottom, 781209, Greiner Bio-One GmbH) were blocked with casein (0.5%, *w/v*) in PBS containing 0.05% (*v/v*) Tween 20 at 4 °C overnight and washed three times with phosphate buffer. A 2-fold dilution series (22 steps) of pure 70S ribosome, 50S or 30S subunit (10 µL per well) in ribosome preparation buffer was prepared in a black 384-well plate. The cf-labeled peptide was dissolved in ribosome preparation buffer (10 µL, 40 nmol/L), added to each well, centrifuged (500 *x g*, 2 min), and incubated at 28 ± 1 °C for 90 min. Fluorescence polarization was measured on a PARADIGM™ microplate reader (Beckman Coulter, Krefeld, Germany) in the top reading position ($\lambda_{exc}$ = 485 nm, $\lambda_{em}$ = 535 nm). Data were fitted to a nonlinear dose-response logistic transition equation [y = y0 + a/(1 + (x/x0)b)] using the Levenberg-Marquardt algorithm, with $K_d$ represented by the x0 coefficients (GraphPad Prism for Windows v. 10.1.2, San Diego, CA, USA).

Inhibition constants ($K_i$) were determined for the pure 70S ribosome using cf-labeled and unlabeled peptides in the ribosome preparation buffer. Black 384-well plates were prepared as described above. Unlabeled peptides or antibiotics (20 µL, 300 µmol/L) were added to ribosome preparation buffer in a 2-fold serial dilution series of 21 steps. Ribosome solution adjusted to a ribosome concentration corresponding to ~90% of the upper $K_d$ plateau obtained for the cf-labeled peptide of interest was added to each well (10 µL). After incubation (28 °C ± 1 °C, 90 min), the cf-labeled peptide (10 µL, 80 nmol/L) was added and incubated (28 °C ± 1 °C, 90 min). Fluorescence polarization was measured on a PARADIGM™ microplate reader in the top reading position ($\lambda_{exc}$ = 485 nm, $\lambda_{em}$ = 535 nm). Data were fitted to a nonlinear, dose-response logistic transition equation [y = y0 + a/(1 + (x/x0)b)] using the

Levenberg-Marquardt algorithm, with the half-maximal inhibition constants ($IC_{50}$) represented by the x0 coefficients (GraphPad Prism for Windows v. 10.1.2, San Diego, CA, U.S.A.). The $K_i$ values were calculated as described by Mathias and Jung[39].

## Cross-link-study with 70S ribosome

Purified 70S ribosome in ribosome preparation buffer was incubated with biotinylated peptide (1:1 isotope labeled peptide) at a molar ratio of 1:10 for 60 min at 28 °C. Photo-crosslinking was performed by UV irradiation ($\lambda$ = 350 nm) with a final dose of 4.5 J/cm$^2$ and stored at -80 °C. Ribosomal RNA was digested with RNase I (Thermo Fisher Scientific, Dreieich, Germany) at 26 µg/U for 2 h at 37 °C before washing out unbound peptide by ultrafiltration (Amicon®Ultra-4 centrifugal filters (Merck Millipore) with ribosome preparation buffer.

## Magnetic bead enrichment

All steps were performed using Protein LoBind tubes (Eppendorf). Biotinylated cross-linked complexes were enriched with streptavidin magnetic beads (PureProteome™ Streptavidin Magnetic Bead System; Merck Millipore)[16]. All wash steps were extended to 15 min, including 1% aqueous acetonitrile containing 0.1% TFA as the final wash step, before eluting the bound proteins with 40% aqueous acetonitrile containing 0.1% TFA. The elution was dried by vacuum rotation and reconstituted in ribosome preparation buffer for further analysis.

## SDS-PAGE and Western blot analysis

Proteins were mixed with 5x sample buffer (0.5% bromophenol blue, 62.5 mmol/L Tris-HCl, pH 6.8, 20% glycerol, 2% sodium dodecyl sulfate (SDS), 5% ß-mercaptoethanol), incubated at 95 °C for 10 min, and separated by SDS-PAGE (T = 16%, C = 2.67 %). Gels were stained with Oriole (Oriole Fluorescent Gel Stain; Bio-Rad Laboratories, Feldkirchen, Germany) for 60 min and Coomassie overnight or blotted onto a LF-PVDF membrane using the TransBlot Turbo RTA Transfer Kit (Bio-Rad Laboratories) at 2.5 V for 7 min. The membrane was incubated with the particle-based immunoassay PIA-Pink technology, which includes washing with PIA-Pink-Block for 5 min and incubation overnight (4 °C, 30 rpm) with PIA-Pink-Mouse (PiNa-Tec, Hamburg, Germany) containing 2 µg of anti-biotin mouse mAb (Life Technologies GmbH, Darmstadt, Germany). The membrane was dried for one hour at room temperature and photographed (ChemiDoc MP CCD camera system, Bio-Rad Laboratories).

## Tryptic in-gel digestion

Bands of interest were manually excised from the gels, cut into small pieces, and washed three times with ammonium bicarbonate buffer (50 mmol/L) containing 30% acetonitrile (v/v) and once with acetonitrile (5 min incubation times). The supernatant was discarded, and the gel pieces were air-dried. Trypsin (0.1 µg in 20 µL of 3 mmol/L ammonium bicarbonate) was added and incubated overnight at 37 °C. The peptide solution was transferred to another tube, and fresh acetonitrile was added to the gel pieces. After five minutes, the solution was combined with the digestion solution and dried under vacuum (60 °C, 1.5 h).

## Mass spectrometry

The tryptic digest was reconstituted in aqueous acetonitrile solution (3% v/v, 20 µL) containing formic acid (0.1% v/v), sonicated (3 min), and centrifuged (13,400× *g*, 15 s). Samples were analyzed on a nanoACQUITY UPLC system (Waters Corp., Eschborn, Germany) coupled online to an electrospray ionization quadrupole-time-of-flight mass spectrometer (Synapt G2-Si MS, Waters). Peptides were loaded in aqueous acetonitrile (3%, v-v) containing formic acid (0.1%, v-v) on a nanoACQUITY UPLC® precolumn (2G-VM Trap 5 µm Symmetry® C18 180 µm × 20 mm column) at a flow rate of 5 µL/min for 6 min and separated on an AQUITY UPLC ® M-Class Peptide BEH C18 column

(100 mm length, internal diameter 75 μm and particle diameter 1.7 μm) using water (eluent A) and acetonitrile (eluent B), which both containing formic acid (0.1% v-v). Analytes were eluted by a linear gradient from 3% to 40% eluent B in 18.5 min and to 95% eluent B in 5.5 min at a flow rate of 0.3 μL/min. Peptides were analyzed in positive ion mode using the following setup: capillary voltage of 3 kV, source temperature of 100 °C, sampling cone voltage of 30 V, desolvation temperature of 250 °C, cone gas flow of 20 L/h, nanoflow gas pressure of 0.2 bar, and purge gas flow of 600 mL/h. Mass spectra were acquired by High Definition Data Directed Analysis (HD-DDA) from $m/z$ 300 to $m/z$ 1800 with a survey scan time of 0.2 s. Tandem mass spectra were triggered with an intensity threshold of 1000 from $m/z$ 50 to $m/z$ 5000 (scan time of 0.5 s) considering the six most intense signals of the survey scan. Collision-induced dissociation (CID) was performed in the trap cell using an energy ramp from 12.3 eV to 17.8 eV at 300 $m/z$ to 51 eV to 70.6 eV at $m/z$ 5000, followed by separation in the IMS cell. The postacquisition lock mass calibration was based on the doubly protonated signal of Glu-1fibrinopeptide B ($m/z$ 785.84206) recorded every 30s.

The acquired data were processed using PEAKS Studio 10.6 Xpro (Bioinformatics Solutions, Waterloo, Canada). Data refinement started with a mass correction, considering a lock mass and an error tolerance of 0.5 Da up to a charge state of 8. De novo processing considered a parent mass error of 20 ppm, a fragment mass error of 80 mDa, up to three missed cleavage sites, trypsin as protease (cut "RK" no_cut"P"), and oxidation (+15.99) or deamidation (+0.98) as variable modifications. Database searches were performed using an *E. coli* database downloaded from uniprot (March 28, 2020) containing 4392 protein sequences and additionally the sequences of Api88, mCherry, mAzamy, trypsin, and streptavidin, with a false discovery rate of 1% at the peptide level. The data were exported as peptides.pep.xml files and loaded to Skyline (64-bit, version 22.2.0.351, MacCoss Lab Software, Washington, US). A spectral library was built from all identified peptides. RAW files were uploaded into Skyline and spectra were compared to the built spectral library. Precursor masses were manually checked in all mass spectra (precursor mass error of 20 ppm, isotope dot product of ≥ 0.9), and no signal or mismatched signals were removed. Proteins identified in both replicates were considered enriched based on the peak area ratios obtained in 70S ribosome samples UV-irradiated in the presence or absence (control) of biotin-SG-Api88(Y7B).

## Preparation of 50S•Api137 and 50S•Api88 complexes for cryo-EM

Purified 50S subunits (0.3 μM) were incubated with Api137 or Api88 (30 μM) in Tico buffer (20 mM HEPES/KOH, pH 7.6, 30 mM AcOK, 6 mM Mg(AcO)₂, 4 mM 2-mercaptoethanol) at room temperature for 20 min. The complexes were applied to glow-discharged (Pelco Easy Glow) holey carbon grids (400 Mesh Cu R2/2, Quantifoil MicroTools GmbH) and plunge-frozen in liquid ethane using a Vitrobot Mark IV (Thermo Fisher Scientific) device.

## Cryo-electron microscopy and data processing

Data for the 50S•Api137 and 50S•Api88 complex were collected on a Titan Krios G3i transmission electron microscope (Thermo Fisher Scientific, Server Version 2.15.3, TIA Version 5.0) operated at an acceleration voltage of 300 kV and equipped with an extra-bright field-emission gun, a BioQuantum post-column energy filter (Gatan) and a K3 direct electron detector (Gatan, Digital Micrograph Version 3.32.2403.0). All images were recorded in low-dose mode as dose-fractionated movies using EPU Version 2.8.1 (Thermo Fischer Scientific) with a maximum image shift of 12 μm using aberration-free image shift. Overall, 360 movies of the 50S•Api137 complex with a total dose of 24.6 e Å−2 each split over 25 fractions (with an individual dose of 0.984 e Å−2 per fraction) were recorded in energy-filtered zeroloss

(Slit Width 20 eV), nano-probe mode at a nominal magnification of ×42,000 (resulting in a calibrated pixel size of 1.03 Å on the specimen level) in super-resolution mode with a 100 μM objective aperture. Data were recorded for 2.3 sec with defocus values ranging from −0.5 to −2 μm.

For the 50S•Api88 complex overall, 708 movies with a total dose of 26 e Å−2 each split over 25 fractions (with an individual dose of 1.04 e Å−2 per fraction) were recorded in energy-filtered zeroloss (Slit Width 20 eV), nano-probe mode at a nominal magnification of ×64,000 (resulting in a calibrated pixel size of 0.667 Å on the specimen level) in super-resolution mode with a 100 μM objective aperture. Data were recorded for 1.47 sec with defocus values ranging from −0.5 to −2 μm.

All preprocessing steps were performed in WARP[40]. Therefore, fractions were binned (Api137: 1.18 Å/pixel, Api88: 0.998 Å/pixel), aligned and dose weighted. CTF estimation was performed using a resolution range of 4-12 Å. A pre-trained 50S model was used as the template for particle picking. For 50S•Api137 complex data processing, particle images were extracted with a box size of 300. If not stated otherwise, Cryosparc v3.1[41] was used for the identification and refinement of final classes. Initial generation of an internal 50S template was achieved using ab-initio classification. All picked particles were aligned to the 50S map using Homogenous refinement. For the isolation of a highly resolved subset, 3D classification was performed in Relion 3.1[42] without alignment. Particles from the best-resolved class were subjected to non-uniform refinement using on-the-fly per-particle and global-CTF parameter correction to achieve a final map at 2.64 Å.

For 50S•Api88 complex data processing, particle images were extracted with a box size of 360. All picked particles were aligned to the initial 50S template map (resampled to 360 box size) using Homogenous refinement. For the isolation of a highly resolved subset, 3D classification was performed in Relion 3.1 without alignment. Particles from the best-resolved class were subjected to non-uniform refinement using on-the-fly per-particle, global-CTF parameter correction. Finally, Ewald sphere correction was performed to achieve a final reconstruction at 2.46 Å.

Focused classifications for all Api88 binding sites were attempted using different parameters (Mask size, regularization parameter (Relion 3.1), class size, starting resolution) and using different softwares (Relion 3.1, Cryosparc 3D variability, Cryosparc 3D classification). Classification mostly resulted in convergence to a single class. Solely a single attempt for the PET using decimated particles (2.66 Å/pixel) resulted in a separation and pronounced Api88 density (Supplementary Fig. 8). Respective class was subjected to a non-uniform Refinement, resulting in a 2.55 Å map. This class, however, still showed features of all modeled conformations.

## Model building and structural analysis

An initial 50S atomic model was taken from a high-resolution 70S refinement (PDB:7K00[43]). A starting model for Api137 was taken from [¹¹(PDB: 5O2R). Initially, 50S and Api137 models were rigid body docked in ChimeraX 1.5[44] in the 50S•Api137 map. For subsequent modelling, unsharpened and sharpened maps were used. Models were combined and adjusted by iterative model building and real-space refinement into the EM-density using Coot 0.9.6[45] and Phenix 1.20[46]. For Api88, the C-terminal carboxylate group was manually adjusted to an amide. The 50S•Api137 model was placed in the 50S•Api88 map, followed by iterative manual and automatic adjustments in Coot and Phenix. Likewise, Api88 models were rigid body docked in the novel binding sites at the tunnel end and within domain 3 and adjusted. Three different models were constructed for the tunnel binding site. Coulombic potentials at different binding sites were calculated using ChimeraX using 4 Å surface models. For the comparison of geometries between the peptides at different binding sites, N-terminal, central and C-terminal segments were rigid body fit in ChimeraX into a molmap at

4 Å of Api137 within the tunnel binding site. ChimeraX was used for visualizations.

**Molecular dynamics (MD) simulations.** The cryo-EM models representing conformations I, II and III of the 50S•Api88 complex were used to obtain starting structures for MD simulations. All residues positioned within 35 Å of the Api88 peptide were extracted from each model. The models did not include ions and structural water molecules. Therefore, a previously resolved 50S structure (PDB ID 6PJ6)[47] was aligned to the 50S•Api88 cryo-EM models and ions and structural water molecules within 5 Å were included in the simulation system. The WHATIF software[48] was used to determine histidine protonation states. The system was then placed at the center of a dodecahedral box, with a minimum distance of 1.5 nm between the atoms and the box boundaries. Solvation with OPC water[49] was carried out using the program solvate. GENION[50] was used to neutralize the system with $K^+$ ions and to add 7 mmol/L $MgCl_2$ and 150 mmol/L KCl. The $K^+$ and $Cl^-$ parameters from Joung and Cheatham[51] and the microMg parameters from Grotz and Schwiertz[52] were used.

Lennard-Jones and short-range electrostatic interactions were computed within a cut-off of 1 nm. Long-range electrostatic interactions were calculated using the particle-mesh Ewald summation for distances larger than 1 nm[53] and with a grid spacing of 0.12 nm. Bond lengths were constrained using the LINCS algorithm[54,55]. Virtual sites[55] were used for hydrogen atoms, allowing a 4-fs integration time step. Solute and solvent atoms were independently coupled to a heat bath at 300 K, using velocity rescaling[56] with a coupling time constant of 0.1 ps. The pressure was coupled to a stochastic cell rescaling barostat[57] with a time constant of 5 ps and nstpcouple was set to 10 steps. All simulations were performed using GROMACS 2023[50] and the amber14sb force field[58].

Energy minimization was performed with harmonic position restraints ($k = 1000$ kJ mol$^{-1}$ nm$^{-1}$) on the solute-heavy atoms. Subsequently, 5 replicas of each starting conformation were simulated. For each replica, the system was equilibrated in two steps. During the first step (0–50 ns), harmonic position restraints ($k = 1000$ kJ mol$^{-1}$ nm$^{-1}$) were applied to all solute-heavy atoms. During the second step (50–70 ns), the position restraints applied to the atoms of residues located further than 25 Å from Api88 were linearly decreased to force constants obtained from full-ribosome simulations[59]. The position restraints on heavy atoms of residues located within 25 Å of Api88 were linearly decreased to zero. Finally, production runs (70–2070 ns) were computed, keeping the position restraints only on the heavy atoms placed in the outer shell. Coordinates were recorded every 5 ps. Prior to all analyses, a rigid-body fit was performed between the phosphate atoms in each individual trajectory frame and the phosphate atoms of the initial structures.

**Conformational modes of Api88.** We obtained the main conformational modes of the Api88 peptide in the PET by performing a principal component analysis[60] (PCA) using the GROMACS tools COVAR and ANAEIG[50]. Frames were taken every nanosecond from all MD production trajectories and concatenated into a single trajectory. We used COVAR on this combined trajectory to calculate the covariance matrix of the 148 heavy atoms of the peptide. The eigenvectors of the covariance matrix with the largest eigenvalues represent the main conformational modes. All trajectories originating from the same starting structure were then concatenated, maintaining the original output timestep of 5 ps. We used ANAEIG to calculate the projection of these MD trajectories onto the two eigenvectors associated with the two largest eigenvalues. These represent the conformational modes that capture most of the variance in the MD trajectory data (54 %). The same program was used to identify the extreme conformational states of the peptide along each conformational mode. A 2D histogram of the projection data was plotted using Python[61] and

the packages Matplotlib[62] and Seaborn[63]. The extreme states along each of the two selected conformational modes were visualized using PyMOL[64].

**Computation of density maps.** The software GROmaρs was used to calculate density maps from MD structures via multi-Gaussian spreading[65]. A section of the 50S•Api88 cryo-EM map, restricted to a region of 5 Å around the atoms of the modeled conformations I-III, was used as a reference map. The specified map region was extracted in ChimeraX[44]. The computed maps have the same grid and resolution as this reference. A Gaussian atomic spread width of $\sigma = 0.11$ nm was used for the map computations. We determined the optimal $\sigma$ by computing density maps of the entire 50S•Api88 model (conformation I) at different atom spread widths ranging from $\sigma = 0.05$ nm to $\sigma = 0.15$ nm and then calculating the correlation coefficients between these maps and the cryo-EM map. An atom spread width of $\sigma = 0.11$ nm yielded the best correlation and was used in the following analyses.

**Comparison of MD ensembles to cryo-EM map.** Ensemble density maps were calculated using GROmaρs as described above. A custom Python script was used to identify the set of optimal weights. The script used the Mrcfile package[66] to parse the ccp4 files produced by GROmaρs, and NumPy[67] to process the map data. The results were visualized using Matplotlib and Seaborn.

**Finding correlation-optimized structure sets from MD trajectories.** To identify structures from the MD simulations that best represent the cryo-EM map, we first obtained a pool of Api88 structures. We discarded the first 500 ns of each trajectory. We then extracted structures at 250 ns intervals from all MD trajectories, resulting in 105 structures. For each structure, we calculated a corresponding density map using GROmaρs. To obtain the optimal set of $N$ structures representing the cryo-EM map, we first randomly selected $N$ structures from the pool. To test how many structures are required, we varied $N$ between 1 and 5. For each $N$, five replica sets with different structures were generated. The set of $N$ structures was used as a starting point for an algorithm that iteratively optimizes the correlation coefficient of the set with the cryo-EM map by randomly replacing the structures with others from the pool. Initially, the algorithm determined the optimal weights of the structures in the set (see above) and stored the correlation with the cryo-EM map. Subsequent iterations consisted of two steps: First, a structure from the set was randomly selected and exchanged by another structure from the pool. Then, the weights were optimized for the new set of structures, and the correlation coefficient was compared with that of the previous iteration. If the new correlation coefficient was higher, it was stored, and the updated set was accepted for use in the next iteration. Otherwise, the updated set and its associated correlation coefficient were discarded, and the algorithm continued with the previous set. For each starting set, the algorithm was allowed to run until the correlation coefficients of all replicas with the same number of structures matched at least to an accuracy of $10^{-2}$. The algorithm was implemented in Python, using NumPy for array computations. Its runtime was greatly reduced by using the just-in-time-compiler Numba[68]. The results were visualized using Seaborn and Matplotlib.

### Reporting summary
Further information on research design is available in the Nature Portfolio Reporting Summary linked to this article.

## Data availability
Cryo-EM density maps and atomic models are stored in EMDB and PDB as follows: EMD-19426, 8RPY (50S in complex with Api137); EMD-19427, 8RPZ (50S in complex with Api88 conf. I); EMD-19428, 8RQ0

(50S in complex with Api88 conf. II); EMD-19429, 8RQ2 (50S in complex with Api88 conf. III).

Mass spectrometry proteomics data have been deposited to panorama with the ProteomeXchange ID PXD044892. Molecular dynamics simulation data are publicly available on zenodo.org: [https://doi.org/10.5281/zenodo.10874716]. All additional data needed to evaluate the conclusions in the paper are present in the paper and/ or the Supplementary Materials. Source data are provided with this paper.

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

## Acknowledgements

We thank Daniela Volke for assistance on mass spectrometric analysis, Stefanie Weinert for expressing TF and determining the dissociations constants of Api88, Api137, and apidaecin 1b to TF, Timo Flügel, Daniel Knappe, Yollete Guillén Schlippe, Norbert Sträter and Franziska Wiechert for scientific discussions. We acknowledge Nora Vázques-Laslop and Alexander Mankin for directing the toeprinting experiments. We also thank the Core Facility for cryo-Electron Microscopy (CFcryoEM) of the Charité - Universitätsmedizin Berlin for support in acquisition (and analysis) of the data. The CFcryoEM was supported by the German Research Foundation (DFG) through grant No. INST 335/588-1 FUGG. This work was funded by Bundesministerium für Bildung und Forschung (BMBF 16GW0300 to C.M.T.S. and 16GW0299K to R.H.) and supported by the Deutsche Forschungsgemeinschaft (DFG) through the cluster of excellence "UniSysCat" (under Germany´s Excellence Strategy-EXC2008/1-390540038 to, C.M.T.S.) and "Multiscale Bioimaging" (EXC 2067/1-390729940 to L.V.B and H.G.). This study was conducted within the Max Planck School Matter to Life, supported by the German Federal Ministry of Education and Research (BMBF) in collaboration with the Max Planck Society (O.B.).

## Author contributions

Data analysis: S.M.L., M.R., O.B., S.G., A.K., J.G. and R.N.; Experimental design: A.K., R.N., C.M.T.S., and R.H.; Funding acquisition: H.G., R.N., C.M.T.S., and R.H.; Sample preparation: S.M.L., M.R., A.K., D.K., and J.G.; Supervision: H.G., L.V.B., A.K., R.N., C.M.T.S., and R.H.; Initial draft: S.M.L., R.H., R.N. and A.K.; Writing: All authors approved the final manuscript.

## Funding

## Competing interests

The authors declare no competing interests.
