## [Peer Review File · Nature Communications]

Multimodal binding and inhibition of bacterial ribosomes by the antimicrobial peptides Api137 and Api88REVIEWER COMMENTS

Reviewer #1 (Remarks to the Author):

In the submitted manuscript, Lauer et al. present the binding mechanisms of two apidaecin antimicrobial peptides to the ribosome to compare their bacterial activity. They applied a number of functional assays to differentiate the activity of both, also comparing to some other AMPs, and then performed cryo-EM structural studies. Understanding the mode of action of antimicrobial peptides is highly important as they offer avenues towards future antibiotics while multidrug resistance is an increasing concern. The authors provide a detailed analysis of the binding modes observed in the cryo-EM studies and offer possible mechanisms resulting from the experimental observations.

While several papers have previously been published on peptides of the apidaecin family, particularly for Api137 (<https://www.nature.com/articles/nsmb.3439>, reference 14, and <https://www.nature.com/articles/s41467-018-05465-1>, reference 6), this is the first cryo-EM structural study on Api88 and Api137. I believe that this study is of relevance to the community as it explores different binding mechanisms to the ribosome, including those independent of release factor. The data are presented in good figures; however, the description currently sometimes misses the reasoning until the concluding sentence of a paragraph. Below is some detailed feedback:

- The title makes the impression that three binding sites apply to AMPs of the apidaecin family in general. However, only two apidaecin peptides were tested and different binding modes were identified. The title should better reflect the conclusion from the study.
- The abstract should be improved to make it easier to follow for readers. It currently jumps from fact to fact without transitions, e.g. it is not mentioned that Api137 is a proline-rich AMP. This can be achieved by using transitions such as: "One such PrAMP is Api137, an ...".
- Lines 66/67 vs. 266/267: "Api137 and Api88 differ only in their C-terminus, i.e., a free acid versus an amide." In contrast to the introduction, the authors later mention additional differences between Api137 and Api88. This should be checked.
- The authors identified one single conformer for Api137 binding to the ribosome, consistent with the literature, but "at least three slightly different binding conformers of Api88". This indicates a lower specificity and leaves the question of which modes are the dominant modes and how relevant the ones captured are in a biological context. This should be discussed.
- Due to the lower specificity of Api88, do the authors believe that the three binding modes presented represent all possible binding modes? Which one is dominant and energetically the most favorable? How could alternative peptides be designed to achieve specific binding into one of these modes?

- In the results section, I believe it would be beneficial if the authors would include an introductory and concluding sentence for the assays to introduce why they did an assay and what the result was. For example, Onc112 is mentioned in line 112 but not introduced at all before. For a publication in a journal with broad readership, the article will be more accessible when this is better contextualized.

- Lines 290-293: "The shifted binding mode and the use of electrostatic interactions instead of hydrogen bonds may indicate an evolutionary "fallback" mechanism to overcome potential resistance mechanisms or to allow inhibition of structurally diverse ribosomes in different bacteria." How did the authors distinguish between the interaction mechanisms? Was this concluded from distances in the cryo-EM models?

- Line 337: The authors refer to unpublished data. The data should either be provided as Supplement, or the statement removed, or it should be cited if it is published elsewhere.

- Line 349: "It was surprising and exciting to explore how the small structural difference at the C-terminus" Changes in charge are known to cause different electrostatic interactions and even secondary structures. Could the authors mention the difference in overall charge and partial charges? Is it really a small change and surprising?

- Line 614: Correct sentence: "An initial 50S atomic model from taken from a high-resolution"

Reviewer #2 (Remarks to the Author):

In the present manuscript, S. Lauer et al. investigated the mechanism of the inhibitory action of two proline-rich AMPs at the bacterial ribosome at the molecular level. The study showed that although Api137 and Api 88 are very similar peptides, differing slightly only at the C-terminus, they bind the ribosome in somewhat different ways. Functional studies and cryo- EM structural studies unambiguously demonstrated the complexity of the binding by deciphering the presence of different conformers. In addition, a second binding site for the two apidaecin analogues near the polypeptide exit tunnel and a third binding site for Api88 were identified. The authors propose a model for the molecular mechanisms of Api137 and Api88 and suggest that the multimodal mode of action of these compounds may make more difficult the development of resistance, an interesting aspect of the development of PrAMPs as antibacterial agents.

The goal of the work is clear, and the experimental design is sound and reasonable. The results provide new insights and new details about the precise mechanism of binding of these peptides to the ribosome.

Due to the large number of figures, some aspects of the reading are quite difficult. For example, figures showing data on the second binding site of Api137 have been included in the main text (Figure 4) and in the case of Api88 in the supplemental materials (extended data, Figure 9). The text indicates that the

binding mode is similar, but the reader cannot readily understand the comparison between the two peptides. In addition: the caption of Figure 4 reports that the data shown refer to both Api137 and Api88, but only data on Api137 are given. Furthermore, do the descriptions on page 9, lines 229-239, refer to Api137, Api88, or both?

In the last sentence of the abstract, the authors state that multimodal mechanisms of action have contributed to the evolutionary success of this diverse group of PrAMPs. Since Api88 and Api137 are artificial peptides, it is unclear what is meant by “evolutionary success” in this context. Do they mean the natural counterparts of the PrMAPs? Regarding the emergence of resistance, could the author cite or report an indication of the frequency of occurrence of bacterial mutants resistant to these PrAMPs?

The results of the cross-linkage mapping of the binding region of Api88 identified some ribosomal proteins uL10, uL23, and uL29 as possible contact sites of Api88. Commentary on these results is lacking, and some of these proteins were not mentioned further in the discussion.

Mode of action of PrAMPs as inhibitory peptides on protein synthesis in bacteria have been performed both in insects and mammals. This aspect should be mentioned in the introduction and in the discussion sections.

Minor points

Page 4 line 78: “P-site tRNA interaction.”

Page 7 line 189: the data given are in extended data figure 6f and not in 5f as indicated

Extended data 2: chloramphenicol is misspelled

Reviewer #3 (Remarks to the Author):

The manuscript by Lauer et al presents a detailed and convincing account of the different modes of binding of two closely related insect PrAMPs (Api137 and -88) with some comparison with the binding of other insect or mammalian PrAMPs. The manuscript is well written and experimental data well supported, although it makes for quite technical reading. In fact, a short section on functional aspects of

the peptides action is followed by a quite detailed description of the structural aspects of binding. It is however difficult to see how this part could be simplified or shortened. It is somewhat assisted by quite effective figures.

There follow some comments and considerations in order of appearance and not necessarily of importance:

Abstract lines 33-35: Given that binding details are based on artificial modified peptides there is a logical jump to the evolutionary success of natural PrAMPs

A general criticism is that the authors assume readers will have a comparable knowledge of the details of ribosomal structure as they have. This is a Communications journal with a very varied readership and some effort should be made to facilitate their reading. For example:

Line 78 is P-tRNA the tRNA in the P site of the ribosome, or tRNA bearing Proline, or possibly the growing peptidyl-RNA?

Line 122: not quite clear

Lines 125-131: is the alteration of Api88 with biotin and photochemical probe completely non-invasive with respect to how it bins to the 70S ribosomes

Line 160: bended  bent

Lines 180 and 190: is it G2586 or U2586?

Line 188 and extended data line 91: Bac7 is about 70 residues. The peptide in ED Fig. 6 appears much shorter. Specify if it is Bac7(1-35), Bac7(1-16) or some other fragment or analog.

Line 230: Unclear which PrAMP is referred to. Maybe should read "Asn3 of Api137 binds..."

Line 244: Again presuming detailed knowledge of ribosomal structure. Should read: "...the peptide is loated above helix H51 and below the phosphate backbone of H49 of 23S rRNA."

Line 247: Ditto. Should read: "...near Arg76 of the large ribosomal subunit uL23 and..."

Line 267: Unclear, should read: "...and two positions substituted with respect to this"

Line 271: "...of the related peptide Onc112 and,..."

Line 328: Again presuming detailed knowledge: It would possibly be useful to explain what TF is and does., also with reference to line332-338. It is interesting that TF is a ribosome-associated chaperone that binds to hydrophobic segments of the nascent chain, such as those recognised by the signal recognition particle. This is not the only chaperone that PrAMPs bind to as Otvos et al reported that apidaecin and other insect PrAMPs bind to it and Scocchi that bovine Bac7(1-35) also does so. It is not clear here if the role of Api137 and -88 is to mask the binding site for TF on the ribosome or to interact with it. The author refer to Api88 binding to TF, and in the same work show it binds also to the other chaperone DnaK (ref. 27). In this respect is there a common aspect of their binding to PrAMPs?

A key aspect of this work is that PrAMPs not only bind to specific binding sites with different possible conformations, but that they bind at different sites along the peptide channel, close to the PTC, close to the PET and just outside it. This begs the question if longer PrAMPs could cover all of these sites, or indeed if it could be possible to have tandem repeat Api peptides, possibly with appropriate spacers to do that.

Reviewer #4 (Remarks to the Author):

The manuscript by Lauer et al., entitled "Antimicrobial peptides of the apidaecin family inhibit the bacterial ribosome by multimodal mechanisms involving three different sites on the large ribosomal subunit" characterizes the inhibitory action of the antimicrobial peptide Api88, an amide derivative of the previously characterized Api137. Binding assays and cryo-EM structure determination compare the binding properties of the two peptides. Using release factor-free ribosomes in in vitro translation assays and binding affinity measurements, the authors establish distinct modes of action for Api137 and Api88. These assays suggested that binding of Api137, and therefore translation inhibition, is enhanced by RF1, but binding and translation inhibition by Api88 is independent of RF1. The structures of the two peptides bound to the 50S subunit show a dynamic binding mode for Api88 bound to the previously known site in the peptide exit tunnel. Furthermore, the authors identify an additional binding site for Api137 and Api88 at the end of the peptide exit tunnel, and a third site for Api88 in domain III of the 50S subunit. In the peptide exit tunnel, the EM density shows that the neutral charge at the C-terminus of Api88, as

opposed to the negative charge in Api137, causes the whole peptide to shift toward the PTC, resulting in dynamic alternate conformations of the Api88 peptide.

The significance of the paper is the identification of a distinct mode of action for the near-identical peptide Api88. However, the structural analysis is preliminary, due to the mixed occupancy of Api88 in the peptide exit tunnel.

This leads to this reviewer's major concern:

The authors did not attempt to perform focused 3D classification to better resolve the alternative conformations of Api88 within the peptide exit tunnel. As this may affect the analysis, this reviewer suggests this should first be attempted. The EM density associated with Api88 is quite weak, and barely visible at site III. Again, focused classification of the particles may improve density by sorting out 50S subunits not containing Api88 at that site.

It is likely that the binding of Api88 at site III (and not Api137) is due to the amide C-terminal end of the peptide. However, the authors do not discuss this at all. Can any detail be drawn from the structure?

In Fig. 1, why not test the binding of Api137 and Api88 to the 30S subunit? Could there be additional binding sites in the 30S subunit?

The translation inhibition assays presented in Fig. 1 would benefit from ribosome toeprinting experiments with Api88 and Api137, by possibly providing clear evidence as to whether Api88 stalls ribosomes at stop codons or not. This would provide further support to the hypothesis regarding an 'evolutionary fall-back mechanism' of inhibition by Api88 put forward by the authors.

Figures are repetitive with incomplete descriptions. Extended Data Fig. 5 and Fig. 2 are giving the same information.

Fig. 3 legend is incomplete. What are the transparent volumes outside the surface charge representations, full volume or volume extractions around the model?

Fig. 4j, is this a rotated view of panel i? Panel h does not seem to be an electrostatic representation.

Fig. 5, please review labels in panels d, e. A1600 is a C, A1348 is a C, A1345 is a C, A1396 is a U. One has to use a very low contour level to visualize “some” density of Api88 at site III. Again, please attempt to better sort particles to improve density.

Extended Data Fig. 6a – significance of such a comparison is unclear, considering the dynamic binding mode of Api88.

Extended Data Fig. 6g – figure crowded with the unnecessary distances, which are also missing description in the legend.

Extended Data Fig. 10 – legend unclear, needs more and better description.

In several instances, the authors are missing important citations. For example, please cite the Steitz group as they also published structures of Onc112 and Bac7 bound to the ribosome (NSMB 2015, NAR 2016). The studies were published back-to-back together with the Innis group in the same journals.

Line 47: “ESKAPE” and not “ESKAPEE”.

Line 75: “bypass” would be better with “readthrough”

Lines 92-94: “First, we established a protocol for the isolation of high-purity E. coli 70S ribosomes [17] uncontaminated by other proteins, including the release factors RF1 and RF2 (Extended Data Fig. 1),...” “we” is not clear as none of the author seem to have participated in the work referenced (ref. 17). Also, how is this purification procedure better to remove contaminant proteins? It is not clear from the Ext. Data Fig. 1.

Line 189: Ext. Data Fig. 5f should be Ext Data Fig. 6f

Lines 208-218: refer to specific panels in Ext. Data Fig. 8

Lines 322 and 324: pi, use the Greek letter instead

Line 328: "...via distinct binding sites the trigger factor (TF)..." relative to the trigger factor?

There are three corresponding authors. Yet, only two email addresses are provided.

On the front page, the star refers to a "present address". However, there is no star in the authors' list.

Reviewer #5 (Remarks to the Author):

Antibiotic resistance is a global phenomenon exacerbated in hospital settings, resulting in increased patient morbidity rates. Development of novel therapeutics that overcome antibiotic resistance are therefore important to circumvent this challenge. Api137 and the C-terminal amine-modified Api88 are peptide therapeutics that have been demonstrated previously to be effective antibiotics, particularly targeting, and inhibiting proper mRNA translation by the 70S ribosome. Lauer S, et al present a combination of translation inhibition, binding affinity, cryo-EM, and mass spectrometric data to better understand the differences in binding of Api137 and Api88 to the 70S and 50S E. coli ribosomal subunits. Using this combinatorial approach, they were able to show differences in the translation inhibition efficiency, binding affinity, and binding modes between Api137 and Api88 and conclude that these differences may explain the differences in their mechanism of action. Understanding the structure-function relationship when designing therapeutics is critical for informed drug design and this work is a valuable contribution to this effort.

Overall, I recommend this paper to be published. Prior to publication, I do have some edits that I believe would help to further improve the clarity and quality of the work presented, which I outline below. Please note that since I am not a cryo-EM expert, I am not able to comment on the validity of the data acquisition, processing, and interpretation presented here, and trust that other reviewers will make sound recommendations based on their expertise in this area.

1. Due to the broad research audience of Nature Communications articles, improvement of clarity of the text to enhance the readers comprehension of the overall significance of the work would be beneficial. For example, many readers may not be experts in AMPs and therefore may not understand why "AMPs attacking bacterial membranes often have low safety margins". Please clarify this in the text with a few words or one sentence to explain this. The overall text is quite well written, and just requires a few minor tweaks like the one described above for minor improvement.

2. The dissociation constants presented should be stressed by the authors to be apparent K_d values since they report 2 binding sites for Api137 and 3 for Api88. With the technique used to measure binding affinity, it is not possible to parse between the binding events and thus these reported K_d values represent an apparent value for a combination of these binding events taking place. This could explain the discrepancy between Api88 having a lower K_d , but being less effective at inhibiting translation.

3. The authors discuss MS data in the main text, have several methods sections about it, but the data is not presented anywhere in either the main text figures or supplemental figures. Inclusion in supplemental material seems appropriate, and making raw and processed files openly available could also be completed for data transparency purposes and is highly encouraged in the mass spec community (can upload data to MassIVE data repository at <https://massive.ucsd.edu/ProteoSAFe/static/massive.jsp> and make the data openly available once the manuscript is published). Below is the content I request the authors to address:

i) Inclusion of the MALDI-TOF-MS and ESI-MS spectra that were used to confirm the intact masses of each peptide species seems appropriate to include in the supplemental material since this information was used to confirm correct peptide synthesis. This should be included for all Api137 and Api88 species, including the modified Api88 used for the crosslinking experiments.

ii) The crosslinked in-gel digestion experiment is only summarized in a table in supplemental in Extended Data Fig 3 c, where the authors present summed peak area for the target proteins. The Western blot makes it clear that some protein species are enriched in the Api88 treated samples compared to controls, but the limited information given in the table does not convincingly suggest the identity of these proteins to be primarily L10, L23, and L29. Identifying the most abundant protein species in each dataset would be valuable, and if comparison between the control vs Api88 treated samples is desired, then comparing the peak area for a particular peptide, rather than the summed peak area, would be more informative. In this case, addition of an example chromatographic peak with overlapping charge states and complementary MS/MS spectrum for the peptide of interest would add useful information, while having access to the raw and processed data files would make it possible for interested readers to analyze the data more deeply if desired. Protein sequence coverage is also something that could be added to the table to help the reader understand the validity of the protein assignments.

iii) Related to the above comment, I wonder if the crosslinked Api88 to peptides originating from the L10, L23, and L29 proteins were found in the data? This would be very convincing that these are the proteins in contact with Api88 and would eliminate much of the ambiguity in the results described in ii). If this is not feasible, please describe why you only search for the non-crosslinked peptides in your analysis within the body of your text (page 5 lines 125-131).

REVIEWER COMMENTS

Reviewer #1 (Remarks to the Author):

In the submitted manuscript, Lauer et al. present the binding mechanisms of two apidaecin antimicrobial peptides to the ribosome to compare their bacterial activity. They applied a number of functional assays to differentiate the activity of both, also comparing to some other AMPs, and then performed cryo-EM structural studies. Understanding the mode of action of antimicrobial peptides is highly important as they offer avenues towards future antibiotics while multidrug resistance is an increasing concern. The authors provide a detailed analysis of the binding modes observed in the cryo-EM studies and offer possible mechanisms resulting from the experimental observations.

1.1 While several papers have previously been published on peptides of the apidaecin family, particularly for Api137 (<https://www.nature.com/articles/nsmb.3439>, reference 14, and <https://www.nature.com/articles/s41467-018-05465-1>, reference 6), this is the first cryo-EM structural study on Api88 and Api137. I believe that this study is of relevance to the community as it explores different binding mechanisms to the ribosome, including those independent of release factor. The data are presented in good figures; however, the description currently sometimes misses the reasoning until the concluding sentence of a paragraph. Below is some detailed feedback:

➤ We are glad that reviewer#1 appreciates the scientific impact of our study and are grateful for the comments and suggestions.

PS: We noticed that, according to Nature Communications formatting requirements, the correct term for figures in the supplement is **not** “Extended Data Fig.” **but rather** “Supplementary Fig.”. In the revised version of the manuscript, we use the term “Supplementary Fig.”.

1.2 The title makes the impression that three binding sites apply to AMPs of the apidaecin family in general. However, only two apidaecin peptides were tested and different binding modes were identified. The title should better reflect the conclusion from the study.

➤ To avoid irritation, we modified title:

Multimodal binding and inhibition of bacterial ribosomes by the antimicrobial peptides Api137 and Api88

1.3 The abstract should be improved to make it easier to follow for readers. It currently jumps from fact to fact without transitions, e.g. it is not mentioned that Api137 is a proline-rich AMP. This can be achieved by using transitions such as: “One such PrAMP is Api137, an ...”.

➤ We improved and shortened the abstract, which now reads (lines 29-38):

“Proline-rich antimicrobial peptides (PrAMPs) inhibit bacterial protein biosynthesis by binding to the polypeptide exit tunnel (PET) near the peptidyl transferase center. Api137, an optimized derivative of honeybee PrAMP apidaecin, inhibits protein expression by additionally trapping release factors (RFs), which interact with stop codons on ribosomes to terminate translation. This study shows that Api137 further occupies a second binding site near the exit of the PET

and can repress translation independently of RF trapping. Api88, a C-terminally amidated (-CONH₂) analog of Api137 (-COOH), binds to the same sites, occupies a third binding pocket and interferes with the translation process presumably without RF-trapping. In conclusion, apidaecin-derived PrAMPs inhibit bacterial ribosomes by multimodal mechanisms caused by minor structural changes and thus represent a promising pool for drug development efforts.”

1.4 Lines 66/67 vs. 266/267: “Api137 and Api88 differ only in their C-terminus, i.e., a free acid versus an amide.” In contrast to the introduction, the authors later mention additional differences between Api137 and Api88. This should be checked.

➤ We thank the reviewer for pointing out this inconsistency and have attempted to describe the differences between Api137 and Api88 unambiguously (lines 79-84):

“Here we have focused on two structurally modified versions of apidaecin 1b (GNNRPVYIPQRPPHPRL), which was originally isolated from the western honeybee (*Apis mellifera*). Substitutions at positions 1 (Gly1Orn) and 10 (Gln10Arg) and N-terminal guanidination significantly improved the antibacterial activities of Api137 (Gu-ONNRPVYIPRPPHPRL-OH; Gu: tetramethylguanidine; O: L-ornithine) and its C-terminally amidated analog Api88 (Gu-ONNRPVYIPRPPHPRL-NH₂) [13].”

The other text section (former lines 266-67) was removed.

1.5 The authors identified one single conformer for Api137 binding to the ribosome, consistent with the literature, but “at least three slightly different binding conformers of Api88”. This indicates a lower specificity and leaves the question of which modes are the dominant modes and how relevant the ones captured are in a biological context. This should be discussed.

➤ We thank reviewer#1 for raising this important point. We have now added the following text section to the discussion (lines 390-94):

“In addition, high resolution cryo-EM reconstructions of purified 50S subunits incubated with Api137 or Api88 confirmed multiple binding sites for both peptides. In this respect the reported K_d values should be considered as apparent K_d values, as they represent the average K_d of at least two or three binding sites of Api137 and Api88, respectively, although there may be additional binding sites not visible in the structures.”

Furthermore, we added a section to the discussion to summarize and compare the modes of action of both Api137 and Api88 (lines 399-420):

“Our cryo-EM and MD data suggest the following modes of action for Api137 and Api88: The positively charged apidaecins are electrostatically attracted to the overall negatively charged ribosome (Fig. 8a). The Arg side chains of the peptides have a dual function. They contribute to the positive net charge, and they can participate in π -interactions with nucleobases. The PET exit has many nucleobases exposed on the surface and available for stacking interactions with the Arg, His15, Tyr7, and Pro residues (binding site 2). Upon release of the nascent chain and following diffusion or the negative net charge of the PET interior, Api137 and Api88 migrate into the tunnel that narrows like a funnel. Beyond the central constriction formed by uL4 and uL22, an array of exposed nucleobases protrudes into the interior of the upper tunnel (Figure 5c & 6). One of these nucleobases, the ultra-conserved A751, appears to be an initial anchor point within the PET. Following the electric charge, the peptides move along the nucleobase rugs, π -interacting with the amino acid side chains. At some point the movement stops for Api137, while Api88 – owing to its stronger net positive charge – continues to migrate further towards the PTC, exhibiting an ensemble of conformations before the forward movement stops

(binding site 1). The different conformations may represent the crawling motions that apidaecins utilize to enter the PET. Due to its stable positioning, the C-terminal carboxyl group of Api137 is able to interact with the RFs and arrest them during termination (Fig. 8b). Api88 remains in a more dynamic metastable state, which allows the dissociation of RFs and P-site tRNA (Fig. 8c). In addition, Api88 can adopt a unique binding site (binding site 3) within domain III of the 23S rRNA (Fig. 8d). It is likely that the negatively charged C-terminus of Api137 hinders passage through the narrow cavity between negatively charged helices H51 and H49 of the 23S rRNA.”

1.6 Due to the lower specificity of Api88, do the authors believe that the three binding modes presented represent all possible binding modes? Which one is dominant and energetically the most favorable? How could alternative peptides be designed to achieve specific binding into one of these modes?

➤ This question goes beyond what we can answer with our current biochemical and cryo-EM based findings (similar to reviewer#4’s point 2). For this reason, we initiated a collaboration with Dr. Helmut Grubmüller’s group at the MPI for Multidisciplinary Sciences (Göttingen, Germany) that has a proven track record in molecular dynamics (MD) simulations of nascent polypeptide chains in the PET. They performed MD simulations with Api88 in an environment mimicking the PET. Ole Berendes, Sara Gabrielli, Helmut Grubmüller and Lars V. Bock were involved in planning and conducting the experiments and hence they were added as co-authors on the current version of the manuscript. We obtained three insights from the MD simulations: (i) Api88 is dynamic and adopts an unusually large ensemble of conformations in the PET, (ii) conformation III contributes strongest to the cryo-EM density and (iii) a set of five structures cover the conformational space described by the MD ensemble (lines 244-70):

“Conformational ensemble of Api88 obtained from MD simulations.

To test if Api88 can indeed adopt metastable conformational states, we performed several extensive all-atom MD simulations of Api88 in the PET initiated from conformations I-III. For each of these conformations, we performed 5 simulations of 2 μ s length each. Projection of the obtained trajectories onto the two dominant modes of motion shows that Api88 explores conformations close to its respective initial conformation in the simulations (Fig. 3a). The ensemble of conformations around conformation III remains far away from the other ensembles, indicating that a transition between these ensembles would take substantially longer than the μ s time scale of the simulations. This transition corresponds to a shift along the tunnel axis (Fig. 2e and Fig. 3a), which is expected to have a large free-energy barrier, since many interactions between the peptide and the PET have to be broken and newly formed.

The observation that Api88 can adopt metastable states raises the question of how much they contribute to the overall conformational ensemble. To address this question, we computed a density map for each conformation (I, II, and III) from the combined trajectories initiated from that conformation. The resulting three density maps were then averaged after assigning weights w_1 , w_2 , and w_3 and the correlation coefficient with the cryo-EM map was calculated (Fig. 3b). The weights that gave rise to the highest correlation coefficient (marked with an ‘x’ in the Figure) showed that all three conformations markedly contributed to the cryo-EM density, with conformation III having the highest weight (0.55). The correlation coefficient calculated for the optimally weighted MD ensemble (0.402) is higher than that of the optimally weighted initial conformations (0.375). This increase after including the dynamics observed in the MD simulations supports the notion that Api88 is indeed dynamic and adopts an unusually large ensemble of conformations in the PET. Next, we addressed the question of how many

structures are sufficient to describe the ensemble. To this end, we selected different numbers of structures from the MD ensemble to obtain the highest correlation coefficient (Fig. 3c). With a set of five structures, the correlation coefficient reached that of the MD ensemble. Notably, these structures are distributed across all sub-ensembles I, II, and III (Fig. 3a), further supporting the notion that the cryo-EM map represents an ensemble of metastable states.”

Fig. 3 Conformational dynamics of Api88 in the exit tunnel. **a**, Trajectories Api88 obtained from MD simulations were projected onto the two dominant conformational modes of Api88 relative to the PET. The probability of observing conformations is shown in greyscale. The initial conformations I, II, and III are indicated by blue, orange, and green circles, respectively. The conformational space explored by the combined simulations from each initial conformation is outlined by colored lines. The cyan backbone pictograms along the two axes visualize the respective conformational changes (arrows). Red circles correspond to the 5 structures that optimize the correlation with the cryo-EM map (compare panel c). **b**, Correlation coefficients ρ between the cryo-EM map and densities calculated from different MD ensembles are shown color coded. The ensembles were generated by assigning weights, w_1 , w_2 , and w_3 , to the trajectories initiated from conformations I, II, and III, respectively. The weights corresponding to the maximum ρ are indicated by an 'x'. **c**, For different numbers of structures, the maximum ρ achieved by optimally selecting structures from the simulations is shown. The ρ values obtained from the initial structures and the full MD ensemble are shown as blue and orange dashed lines, respectively.

1.7 In the results section, I believe it would be beneficial if the authors would include an introductory and concluding sentence for the assays to introduce why they did an assay and what the result was. For example, Onc112 is mentioned in line 112 but not introduced at all before. For a publication in a journal with broad readership, the article will be more accessible when this is better contextualized.

➤ We are grateful to the reviewer for this suggestion. To better contextualize the results section, we introduced i) new section headings and both ii) introductory (explaining the motivation) and iii) concluding sentences as follows:

(i) Section headings:

Line 108: Api137 and Api88 suppress ribosomal translation activity.

Line 133: Api137 and Api88 bind ribosomes with different affinities.

Line 171: UV-activatable Api88 forms cross-links with ribosomal proteins.

Line 195: Cryo-EM reveals different ribosome binding properties of Api137 and Api88.

Line 243: Conformational ensemble of Api88 obtained from MD simulations.

Line 272: Altered electrostatics change the interaction sites of Api88 within the 50S subunit PET.

(ii) introductory sentences:

a) Lines 116-18: We first investigated their effect on protein translation in the presence or absence of the release factors using an *in vitro* transcription-translation (ITT) assay [19, 20].

b) Lines 123-25: To further explore the translation inhibition of Api88, the effects of Api88 on ribosome progression were tested by toeprinting, which allows mapping of the site of inhibitor-induced ribosome stalling during *in vitro* translation [21].

c) Lines 137-39: "...Therefore, we used an additional chromatographic purification to obtain highly pure *E. coli* 70S ribosomes (Supp. Fig. 1) [24] and thus more reliable 70S ribosome binding constants. The ribosome binding was studied by a fluorescence polarization assay using 5(6)-carboxyfluorescein (cf)-labeled PrAMPs..."

d) Lines 172-74: "Next, we tested whether Api88 interacts directly with ribosomal proteins. Thus, Tyr7 in Api88 was substituted with 4-benzoyl-L-phenylalanin, which allows UV-induced cross-linking to nearby residues of bound proteins."

e) Line 196-97: "To further interrogate the molecular mechanism of Api88, we incubated purified 50S subunits with Api137 or Api88 and subjected the samples to cryo-EM analysis."

f) Lines 244-45: "To test if Api88 can indeed adopt metastable conformational states, we performed several extensive all-atom MD simulations of Api88 in the PET initiated from conformations I-III."

g) Lines 285-87: "The C-terminally amidated Api88 showed vastly different conformations compared to Api137, which is most likely due to the loss of the negative charge of the C-terminus leading to different electrostatic interactions within the PET (Fig. 4)."

(iii) Concluding sentences:

a & b) Lines 129-31: "...corresponding to ribosomes arrested at the UAG stop codon (Fig. 1d). This was consistent with the results of *in vitro* sfGFP translation (Fig. 1a-c), suggesting that Api88 does not arrest ribosomes about to release the mRNA at the stop codon."

c) Lines 165-69: "... Taken together, Api137 and Api88 have a comparable level of antimicrobial activity..."

d) Lines 191-93: "This suggests either a different or an additional binding site for Api88, although it cannot be excluded that the 4-benzoyl-L-phenylalanine residue is not in close proximity to uL22 and uL4."

e) Lines 241-42: “This suggests that the narrow region near the PTC is the key determinant and regulator of all binding modes [*comment: with respect to binding site 1 within the PET*].”

f) Lines 269-70: “Notably, these structures are distributed across all sub-ensembles I, II, and III (Fig. 3a), further supporting the notion that the cryo-EM map represents an ensemble of metastable states.”

g) Lines 294-97: “In conclusion, while all residues of Api137 reported to interact with the 50S subunit’s PET are still present in Api88, C-terminal amidation appears to change the general binding mode, suggesting that the C-terminal electrostatics are important for positioning of the peptide.”

➤ The PrAMP Onc112 is now introduced in the general introduction (lines 70-73): “Mechanistically, PrAMPs can be divided into two classes based on their effect on the bacterial ribosome [5,8]. First, oncocin-type PrAMPs, such as Onc112, bind to the PET in reverse orientation compared to the nascent protein chain synthesized on the ribosome, thereby blocking the PET and stalling protein translation at the initiation of translation [9]. ...”

1.8 Lines 290-293: “The shifted binding mode and the use of electrostatic interactions instead of hydrogen bonds may indicate an evolutionary “fallback” mechanism to overcome potential resistance mechanisms or to allow inhibition of structurally diverse ribosomes in different bacteria.” How did the authors distinguish between the interaction mechanisms? Was this concluded from distances in the cryo-EM models?

➤ The evolutionary mechanism is an important point that was also critically discussed by other reviewers (see **reviewer#2 point 2** and **reviewer#3 point 1**). In addition, **reviewer#1 (point 3)** suggested to revise the abstract in general. Overall, these hints and comments motivated us to remove the evolutionary aspects from both the abstract and the discussion.

1.9 Line 337: The authors refer to unpublished data. The data should either be provided as Supplement, or the statement removed, or it should be cited if it is published elsewhere.

➤ We thank the reviewer for this hint and have added a reference for the data.

Lines 452-59: “Interestingly, TF was reported as a potential interaction partner of Api88 besides ribosomal proteins and other proteins including DnaK [^{16,25}], based on the above-mentioned cross-linking experiments using Api88(Y7B) [²⁵]. However, studies using fluorescence polarization assays could not confirm that Api88, Api137 or apidaecin 1b bind to recombinant TF ($K_d > 100 \mu\text{mol/L}$) [³⁷]. In the context of the new structures, this data may support the *in vivo* relevance of the identified second and third binding sites to disturb TF-ribosome binding, as Api88 can occupy part of the TF-ribosome contact area represented by uL23 and uL29.”

1.10 Line 349: “It was surprising and exciting to explore how the small structural difference at the C-terminus” Changes in charge are known to cause different electrostatic interactions and even secondary structures. Could the authors mention the difference in overall charge and partial charges? Is it really a small change and surprising?

➤ The reviewer's question is very valid. Per se, it is the smallest possible change in charge that can occur i.e., one subatomic particle (proton) more or less. The net charge changes from +5 (Api137) to +6 (Api88). The effect was surprising since the C-terminal carboxylate was not described to be involved in any 50S interactions (it interacts with the P-Site tRNA in the 70S context). All residues that were previously described to form interactions are still present in Api88. Yet, the C-terminal modification appears to overrule these interactions and highlights the importance of the C-terminal region for positioning of the peptide (lines 285-90):
“The C-terminally amidated Api88 showed vastly different conformations compared to Api137, which is most likely due to the loss of the negative charge of the C-terminus leading to different electrostatic interactions within the PET (Fig. 4). Amidation abolishes the only negatively charged group increasing the overall net charge of the peptide from +5 (Api137) to +6 (Api88). The negatively charged carboxylate close to the PTC could repel Api137 from further protrusion (Fig. 4a).”

1.11 Line 614: Correct sentence: “An initial 50S atomic model from taken from a high-resolution”

➤ Line 759: “We have corrected the sentence to “An initial 50S atomic model **was** taken from a high-resolution”

We thank reviewer#1 for the critical and constructive review.

Reviewer #2 (Remarks to the Author):

In the present manuscript, S. Lauer et al. investigated the mechanism of the inhibitory action of two proline-rich AMPs at the bacterial ribosome at the molecular level. The study showed that although Api137 and Api 88 are very similar peptides, differing slightly only at the C-terminus, they bind the ribosome in somewhat different ways. Functional studies and cryo- EM structural studies unambiguously demonstrated the complexity of the binding by deciphering the presence of different conformers. In addition, a second binding site for the two apidaecin analogues near the polypeptide exit tunnel and a third binding site for Api88 were identified. The authors propose a model for the molecular mechanisms of Api137 and Api88 and suggest that the multimodal mode of action of these compounds may make more difficult the development of resistance, an interesting aspect of the development of PrAMPs as antibacterial agents.

The goal of the work is clear, and the experimental design is sound and reasonable. The results provide new insights and new details about the precise mechanism of binding of these peptides to the ribosome.

➤ We are glad to hear that reviewer#2 appreciates our work and the details we provide about the precise binding mechanism of the apidaecin peptides to the ribosome.

PS: We noticed that, according to Nature Communications formatting requirements, the correct term for figures in the supplement is **not** “Extended Data Fig.” **but rather** “Supplementary Fig.”. In the revised version of the manuscript, we use the term “Supplementary Fig.”.

2.1 Due to the large number of figures, some aspects of the reading are quite difficult. For example, figures showing data on the second binding site of Api137 have been included in the main text (Figure 4) and in the case of Api88 in the supplemental materials (extended data, Figure 9). The text indicates that the binding mode is similar, but the reader cannot readily understand the comparison between the two peptides. In addition: the caption of Figure 4 reports that the data shown refer to both Api137 and Api88, but only data on Api137 are given. Furthermore, do the descriptions on page 9, lines 229-239, refer to Api137, Api88, or both?

➤ We agree that the direct comparison of the Api137 and Api88 models corresponding to binding site 2 is useful. Therefore, we added Supp. Fig. 12d where the two models can be compared side by side.

Moreover, we have removed Api88 from the main Fig. 4 (now Fig. 5) caption. “Additional binding site of Api137 at the PET exit.”

The text passage line 229 following now corresponds to line 320 following. The wording was changed and now the text clearly refers to Api137:

“Binding of Api137 was mediated by multiple hydrogen bonds and stacking interactions (Fig. 5 e-g)...”

2.2 In the last sentence of the abstract, the authors state that multimodal mechanisms of action have contributed to the evolutionary success of this diverse group of PrAMPs. Since Api88 and Api137 are artificial peptides, it is unclear what is meant by “evolutionary success” in this context. Do they mean the natural counterparts of the PrMAPs? Regarding the emergence of resistance, could the author cite or report an indication of the frequency of occurrence of bacterial mutants resistant to these PrAMPs?

➤ The evolutionary mechanism is an important point that has been viewed critically also by other reviewers (see **reviewer#1 point 8** and **reviewer#3 point 1**). Overall, these hints and comments motivated us to remove the evolutionary aspects from both the abstract and the discussion.

2.3 The results of the cross-linkage mapping of the binding region of Api88 identified some ribosomal proteins uL10, uL23, and uL29 as possible contact sites of Api88. Commentary on these results is lacking, and some of these proteins were not mentioned further in the discussion.

➤ We agree with the reviewer that the description and interpretation of the x-link experiments needed improvement. Reviewer#5 expressed a similar criticism (point 3iii). We addressed these issues by reorganizing the results section **UV-activatable Api88 forms cross-links with ribosomal proteins**. The relevant section extends from lines 186-93.

“...The corresponding bands were excised from the gel, incubated with trypsin, and analyzed by mass spectrometry, which identified the ribosomal proteins uL10, uL23, and uL29 of the large subunit as likely contact sites for biotin-SG-Api88(Y7B) (Supplementary Fig. 3, 4 and 5). However, the peptides from uL10 were detected with low signal intensities. In contrast to this, a previous study of Api137 identified interactions with uL22 and uL4 using a genetic screen [10]. This suggests either a different or an additional binding site for Api88, although it cannot be excluded that the 4-benzoyl-L-phenylalanine residue is not in close proximity to uL22 and uL4.”

The proteins in question are now also discussed in Supp. Figure 4.

2.4 Mode of action of PrAMPs as inhibitory peptides on protein synthesis in bacteria have been performed both in insects and mammals. This aspect should be mentioned in the introduction and in the discussion sections.

➤ We also consider the points listed by the reviewer to be very important and are now addressing them in the revised introduction (lines 62-65):

“...This includes insect-derived proline-rich AMPs (PrAMPs), which represent promising lead structures because humans, unlike other mammals, are unable to produce them as part of innate immunity and thus appear to have a gap in their antibacterial repertoire that could be filled therapeutically [5]. ...”

And lines 70-78:

“...Mechanistically, PrAMPs can be divided into two classes based on their effect on the bacterial ribosome [5,8]. First, oncocin-type PrAMPs, such as Onc112, bind to the PET in reverse orientation compared to the nascent protein chain synthesized on the ribosome, thereby blocking the PET and stalling protein translation at the initiation of translation [9]. Second, Api137, a representative of apidaecin-type PrAMPs, binds to the PET of the large ribosomal subunit near the peptidyl transferase center (PTC) in the same orientation as the nascent protein chain. It inhibits translation termination by trapping the release factors RF1 and RF2, which arrests translating ribosomes at the stop codon or promotes stop codon readthrough, resulting in the expression of C-terminally elongated proteins [10-12]. ...”

Minor points

2.5 Page 4 line 78: “P-site tRNA interaction.”

➤ We have corrected it accordingly (lines 91-93):
“...as the interaction of the C-terminal carboxyl group of Api137 with the P-site tRNA...”

2.6 Page 7 line 189: the data given are in extended data figure 6f and not in 5f as indicated

➤ We have corrected it accordingly and changed Supp. Fig. **5f** to now **9f**. See also reviewer#4 point 17.

2.7 Extended data 2: chloramphenicol is misspelled

➤ We have corrected the spelling of chloramphenicol.

We thank reviewer#2 for the critical and constructive review.

Reviewer #3 (Remarks to the Author):

The manuscript by Lauer et al presents a detailed and convincing account of the different modes of binding of two closely related insect PrAMPs (Api137 and -88) with some comparison with the binding of other insect or mammalian PrAMPs. The manuscript is well written and experimental data well supported, although it makes for quite technical reading. In fact, a short section on functional aspects of the peptides action is followed by a quite detailed description of the structural aspects of binding. It is however difficult to see how this part could be simplified or shortened. It is somewhat assisted by quite effective figures.

➤ We are glad that reviewer#3 finds our paper well written and by experimental data well supported.

PS: We noticed that, according to Nature Communications formatting requirements, the correct term for figures in the supplement is **not** “Extended Data Fig.” **but rather** “Supplementary Fig.”. In the revised version of the manuscript, we use the term “Supplementary Fig.”.

There follow some comments and considerations in order of appearance and not necessarily of importance:

3.1 Abstract lines 33-35: Given that binding details are based on artificial modified peptides there is a logical jump to the evolutionary success of natural PrAMPs

➤ The evolutionary mechanism is an important point that was also critically discussed by other reviewers (see **reviewer#1 point 8** and **reviewer#2 point 2**). In addition, **reviewer#1 (point 3)** suggested to revise the abstract in general. Overall, these hints and comments motivated us to remove the evolutionary aspects from both the abstract and the discussion.

3.2 A general criticism is that the authors assume readers will have a comparable knowledge of the details of ribosomal structure as they have. This is a Communications journal with a very varied readership and some effort should be made to facilitate their reading. For example:

➤ The authors are grateful for this remark and made changes as follows:

3.2.1 Line 78 is P-tRNA the tRNA in the P site of the ribosome, or tRNA bearing Proline, or possibly the growing peptidyl-RNA?

➤ We have corrected it accordingly (line 91-93):
“...as the interaction of the C-terminal carboxyl group of Api137 with the P-site tRNA ...”

3.2.2 Line 122: not quite clear

➤ We have rephrased the sentence and hope that its meaning is now clear (lines 165-67):

“Taken together, Api137 and Api88 have a comparable level of antimicrobial activity [17], but the inhibitory effect of Api88 on translating 70S ribosomes is largely independent of the ribosome release factor RF1 and ~50% weaker than for Api137 in the presence of RF1.”

3.2.3 Lines 125-131: is the alteration of Api88 with biotin and photochemical probe completely non-invasive with respect to how it binds to the 70S ribosomes

➤ We are grateful to the reviewer for raising this important point. Actually, these important controls were performed in a previous study (Volke et al., 2015 [25]). The MIC showed the same value, and the K_d showed no significant change:

	MIC	[$\mu\text{g/mL}$]	KD [$\mu\text{mol/L}$]	
Bio-SG-Api88 Y7B	2		n.d.	
Api88	2		n.d.	
cf-SG-Api88 Y7B	8		1.3	+/-0.4
cf-SG-Api88	16		5.0	+/-1.2

(table taken from Volke et al., 2015)

We added to the following text (lines 177-78):

“Previous studies showed similar minimum inhibitory activities and binding constants for biotin-SG-Api88(Y7B) and Api88 [25], suggesting similar behaviors.”

3.2.4 Line 160: bended  bent

➤ We have corrected it (lines 226-27): Conformation II similarly showed a ~~bended~~ C-terminus.

3.2.5 Lines 180 and 190: is it G2586 or U2586?

➤ We thank the reviewer for pointing out this inconsistency. Due to rearranging the text, the nucleobase in question only appears once. We have corrected the text as follows (lines 277-80):

“... mostly via stacking of solvent-facing nucleobases (U2504 in conformation I and II, G2505 in conformation I, U2586 in conformation III...”

3.2.6 Line 188 and extended data line 91: Bac7 is about 70 residues. The peptide in ED Fig. 6 appears much shorter. Specify if it is Bac7(1-35), Bac7(1-16) or some other fragment or analog.

➤ We thank the reviewer for bringing up this important detail. We have now specified that we compared the Api88 conformations to Bac7(1-16). Bac7(1-16) exhibited similar structural features in residues Pro13-Arg14-Pro15 compared to conformation III residues Pro11-Arg12-Pro13 (Supp. Fig. 65f.). We have adapted the manuscript accordingly (lines 422-25):

“Remarkably, all conformations of Api88 were found to be more similar to Onc112 [9, 26] and Bac7(1-16) [27,28] than to Api137, particularly in the positioning of Pro11-Arg12-Pro13, which was comparable to Pro10-Arg11-Pro12 in Onc112 (Supplementary Fig. 9c-e) or Pro13-Arg14-Pro15 in Bac7(1-16) (Supplementary Fig. 9f).”

3.2.7 Line 230: Unclear which PrAMP is referred to. Maybe should read “Asn3 of Api137 binds...”

➤ We are sorry for the confusion and have adapted this whole results section. We now first describe the binding behavior of Api137, followed by a description of how Api88 binds to the 50S subunit.

In addition, it now reads (line 321): “Asn3 of Api137 binds...”

3.2.8 Line 244: Again presuming detailed knowledge of ribosomal structure. Should read: "...the peptide is located above helix H51 and below the phosphate backbone of H49 of 23S rRNA."

➤ We agree and optimized the wording as follows (lines 351-53): "The well resolved central region of the peptide is located above helix H51 and below the phosphate backbone of helix H49 of the 23S rRNA (now **Fig. 7d-f**)"

3.2.9 Line 247: Ditto. Should read: "...near Arg76 of the large ribosomal subunit uL23 and..."

➤ We introduced the ribosomal proteins uL10, uL23 and uL29 first in line 188 and refer to them several times as ribosomal proteins. We assume that the reader has internalized the fact that these are ribosomal proteins. Also, in order not to upset readers familiar with the nomenclature, we do not want to use the term "ribosomal proteins" on every further occasion.

3.2.10 Line 267: Unclear, should read: "...and two positions substituted with respect to this"

➤ We thank reviewer#3 for the suggestion, which is similar to point 4 of reviewer#1. Hence, we removed the text section in the discussion and added a more detailed description in the introduction (lines 79-84):

"Here we have focused on two structurally modified versions of apidaecin 1b (GNNRPVYIPQRPPHPRL), which was originally isolated from the western honeybee (*Apis mellifera*). Substitutions at positions 1 (Gly1Orn) and 10 (Gln10Arg) and N-terminal guanidination significantly improved the antibacterial activities of Api137 (Gu-ONNRPVYIPQRPPHPRL-OH; Gu: tetramethylguanidine; O: L-ornithine) and its C-terminally amidated analog Api88 (Gu-ONNRPVYIPQRPPHPRL-NH₂) [13]."

3.2.11 Line 271: "...of the related peptide Onc112 and..."

➤ We understand what the reviewer is getting at. Due to rearrangements in the text the family of PrAMPs -including Onc112- is now presented already in the introduction (lines 70-72): "...Mechanistically, PrAMPs can be divided into two classes based on their effect on the bacterial ribosome [5,8]. First, oncocin-type PrAMPs, such as Onc112, bind to the PET in reverse orientation..."

3.2.12 Line 328: Again presuming detailed knowledge: It would possibly be useful to explain what TF is and does., also with reference to line332-338. It is interesting that TF is a ribosome-associated chaperone that binds to hydrophobic segments of the nascent chain, such as those recognised by the signal recognition particle. This is not the only chaperone that PrAMPs bind to as Otvos et al reported that apidaecin and other insect PrAMPs bind to it and Scocchi that bovine Bac7(1-35) also does so. It is not clear here if the role of Api137 and -88 is to mask the binding site for TF on the ribosome or to interact with it. The author refer to Api88 binding to TF, and in the same work show it binds also to the other chaperone DnaK (ref. 27). In this respect is there a common aspect of their binding to PrAMPs?

➤ We thank reviewer#3 for raising this point that obviously requires clarification. Apidaecin 1b, Api88, and Api137 do not appear to bind to TF, as fluorescence polarization studies

suggested K_d values above 100 $\mu\text{mol/L}$ (exact values are not available, as the upper plateau was not reached up to the highest protein solubility). We now introduce trigger factor in line 446 as "...molecular chaperone..."

In addition, we hypothesize that Api88 binds to the ribosome within the contact area of TF, which interacts with uL23 and uL29, two proteins we report here as cross-linked to Api88. Apidaecin 1b, Api88, and Api137 bind to DnaK with K_d values of $\sim 5 \mu\text{mol/L}$ and DnaK was originally identified as bacterial target of PrAMPs. We later showed that the major lethal target is the ribosome and thus have not discussed DnaK in detail. However, we have added some comments on DnaK-apidaecin interactions to clarify this aspect (lines 452-59):

"Interestingly, TF was reported as a potential interaction partner of Api88 besides ribosomal proteins and other proteins including DnaK [^{16,25}], based on the above-mentioned cross-linking experiments using Api88(Y7B) [²⁵]. However, studies using fluorescence polarization assays could not confirm that Api88, Api137 or apidaecin 1b bind to recombinant TF ($K_d > 100 \mu\text{mol/L}$) [³⁷]. In the context of the new structures, this data may support the *in vivo* relevance of the identified second and third binding sites to disturb TF-ribosome binding, as Api88 can occupy parts of the TF-ribosome contact area represented by uL23 and uL29."

3.3 A key aspect of this work is that PrAMPs not only bind to specific binding sites with different possible conformations, but that they bind at different sites along the peptide channel, close to the PTC, close to the PET and just outside it. This begs the question if longer PrAMPs could cover all of these sites, or indeed if it could be possible to have tandem repeat Api peptides, possibly with appropriate spacers to do that.

➤ This is an interesting idea. Actually, we tested apidaecin and oncocin bridged peptides using 1 to 20 ethylene glycol (EG) units to obtain flexible spacers of different lengths between both peptides [<https://doi.org/10.1002/psc.2905>]. Onc112-EG₁-Api137 and Onc112-EG₅-Api137 were more active, but most likely due to a better uptake independent of SbmA. Interestingly, the ribosome binding was mostly determined by the C-terminal PrAMP, i.e., the K_d values of Onc112-EG_n-Api137 and Api137 were similar and the K_d values of Api137-EG_n- Onc112 and Onc112 were similar. We are currently investigating other hybrid peptides and the new structures will guide our efforts, but we do not wish to disclose our approach at this time.

We thank reviewer#3 for the critical and constructive review.

Reviewer #4 (Remarks to the Author):

The manuscript by Lauer et al., entitled “Antimicrobial peptides of the apidaecin family inhibit the bacterial ribosome by multimodal mechanisms involving three different sites on the large ribosomal subunit” characterizes the inhibitory action of the antimicrobial peptide Api88, an amide derivative of the previously characterized Api137. Binding assays and cryo-EM structure determination compare the binding properties of the two peptides. Using release factor-free ribosomes in in vitro translation assays and binding affinity measurements, the authors establish distinct modes of action for Api137 and Api88. These assays suggested that binding of Api137, and therefore translation inhibition, is enhanced by RF1, but binding and translation inhibition by Api88 is independent of RF1. The structures of the two peptides bound to the 50S subunit show a dynamic binding mode for Api88 bound to the previously known site in the peptide exit tunnel. Furthermore, the authors identify an additional binding site for Api137 and Api88 at the end of the peptide exit tunnel, and a third site for Api88 in domain III of the 50S subunit. In the peptide exit tunnel, the EM density shows that the neutral charge at the C-terminus of Api88, as opposed to the negative charge in Api137, causes the whole peptide to shift toward the PTC, resulting in dynamic alternate conformations of the Api88 peptide.

➤ We thank reviewer #4 for this accurate summary of our results and the detailed review of our results. We believe that the suggested changes and corrections substantially improved our manuscript.

PS: We noticed that, according to Nature Communications formatting requirements, the correct term for figures in the supplement is **not** “Extended Data Fig.” **but rather** “Supplementary Fig.”. In the revised version of the manuscript, we use the term “Supplementary Fig.”.

4.1 The significance of the paper is the identification of a distinct mode of action for the near-identical peptide Api88. However, the structural analysis is preliminary, due to the mixed occupancy of Api88 in the peptide exit tunnel.

This leads to this reviewer’s major concern:

4.2 The authors did not attempt to perform **focused 3D classification** to better resolve the alternative conformations of Api88 within the peptide exit tunnel. As this may affect the analysis, this reviewer suggests this should first be attempted. The EM density associated with Api88 is quite weak, and barely visible at site III. Again, focused classification of the particles may improve density by sorting out 50S subunits not containing Api88 at that site.

➤ We thank the reviewer for this valuable suggestion. Indeed, we have tested a multitude of different approaches for focused classification for the different binding sites: 1) Relion 3D classification skipping alignment 2) Cryosparc 3D classification and 3) Cryosparc 3D variability analysis. We also tested a variety of different parameters (Mask size, T regularization parameter, different initial/final resolutions, nr. of classes). In most cases, convergence to a single class was observed. For the tunnel binding site, one classification resulted in a slightly more pronounced Api88 density within the tunnel. However, this subset still showed features of all conformations. We concluded that focused classification did not lead to any improvement and decided to mainly use globally sorted maps for interpretations. We are not aware of any other study, which successfully applied focused classification for such a small target (18 amino acids). Especially concerning the PET binding site of Api88, conformations are highly similar and differences would appear only at high resolution.

We have added Supp. Fig. 8 showing the focused classification approach and have added passages in the main text (lines 216-20):

“Focused 3D classifications using different PET masks were performed to test whether the different apparent conformations could be separated into distinct subclasses (see Methods, Supplementary, Fig. 8). While most attempts converged on a single class, one attempt resulted in a more pronounced Api88 density. However, this subclass still indicated features of all modeled conformations in a subsequent refinement.”

And methods section (lines 751-57):

“Focused classifications for all Api88 binding sites were attempted using different parameters (Mask size, regularization parameter (Relion 3.1), class size, starting resolution) and using different softwares (Relion 3.1, Cryosparc 3D variability, Cryosparc 3D classification). Classification mostly resulted in convergence to a single class. Solely a single attempt for the PET using decimated particles (2.66 Å/pixel) resulted in a separation and pronounced Api88 density (Supplementary Fig. 8). Respective class was subjected to a non-uniform Refinement, resulting in a 2.55 Å map. This class, however, still showed features of all modeled conformations.”

Nevertheless, we took reviewer#4’s criticism very seriously -especially since it is similar to reviewer#1’s point 6- and initiated a collaboration with the group of Dr. Helmut Grubmüller (MPI for Multidisciplinary Sciences, Göttingen, Germany). The group has a proven track record in molecular dynamics (MD) simulations of nascent polypeptide chains in the PET. They performed MD simulations with Api88 in an environment mimicking the PET. Ole Berendes, Sara Gabrielli, Helmut Grubmüller and Lars V. Bock were involved in planning and conducting the experiments and hence they appear as co-authors on the current version of the manuscript. The MD data support that our Api88 cryo-EM maps indeed represent an ensemble of metastable states (lines 244-70):

“Conformational ensemble of Api88 obtained from MD simulations.

To test if Api88 can indeed adopt metastable conformational states, we performed several extensive all-atom MD simulations of Api88 in the PET initiated from conformations I-III. For each of these conformations, we performed 5 simulations of 2 μ s length each. Projection of the obtained trajectories onto the two dominant modes of motion shows that Api88 explores conformations close to its respective initial conformation in the simulations (Fig. 3a). The ensemble of conformations around conformation III remains far away from the other ensembles, indicating that a transition between these ensembles would take substantially longer than the μ s time scale of the simulations. This transition corresponds to a shift along the tunnel axis (Fig. 2e and Fig. 3a), which is expected to have a large free-energy barrier, since many interactions between the peptide and the PET have to be broken and newly formed.

The observation that Api88 can adopt metastable states raises the question of how much they contribute to the overall conformational ensemble. To address this question, we computed a density map for each conformation (I, II, and III) from the combined trajectories initiated from that conformation. The resulting three density maps were then averaged after assigning weights w_1 , w_2 , and w_3 and the correlation coefficient with the cryo-EM map was calculated (Fig. 3b). The weights that gave rise to the highest correlation coefficient (marked with an ‘x’ in the Figure) showed that all three conformations markedly contributed to the cryo-EM density, with conformation III having the highest weight (0.55). The correlation coefficient calculated for the optimally weighted MD ensemble (0.402) is higher than that of the optimally weighted

initial conformations (0.375). This increase after including the dynamics observed in the MD simulations supports the notion that Api88 is indeed dynamic and adopts an unusually large ensemble of conformations in the PET. Next, we addressed the question of how many structures are sufficient to describe the ensemble. To this end, we selected different numbers of structures from the MD ensemble to obtain the highest correlation coefficient (Fig. 3c). With a set of five structures, the correlation coefficient reached that of the MD ensemble. Notably, these structures are distributed across all sub-ensembles I, II, and III (Fig. 3a), further supporting the notion that the cryo-EM map represents an ensemble of metastable states.”

Fig. 3 Conformational dynamics of Api88 in the exit tunnel. **a**, Trajectories Api88 obtained from MD simulations were projected onto the two dominant conformational modes of Api88 relative to the PET. The probability of observing conformations is shown in greyscale. The initial conformations I, II, and III are indicated by blue, orange, and green circles, respectively. The conformational space explored by the combined simulations from each initial conformation is outlined by colored lines. The cyan backbone pictograms along the two axes visualize the respective conformational changes (arrows). Red circles correspond to the 5 structures that optimize the correlation with the cryo-EM map (compare panel c). **b**, Correlation coefficients ρ between the cryo-EM map and densities calculated from different MD ensembles are shown color coded. The ensembles were generated by assigning weights, w_1 , w_2 , and w_3 , to the trajectories initiated from conformations I, II, and III, respectively. The weights corresponding to the maximum ρ are indicated by an 'x'. **c**, For different numbers of structures, the maximum ρ achieved by optimally selecting structures from the simulations is shown. The ρ values obtained from the initial structures and the full MD ensemble are shown as blue and orange dashed lines, respectively.

4.3 It is likely that the binding of Api88 at site III (and not Api137) is due to the amide C-terminal end of the peptide. However, the authors do not discuss this at all. Can any detail be drawn from the structure?

➤ We agree that Api88 can most likely enter the third binding site because of the neutral charge at the C-terminus. We have added this speculation to the discussion (lines 417-20):

“In addition, Api88 can adopt a unique binding site (binding site 3) within domain III of the 23S rRNA (Fig. 8d). The negatively charged C-terminus of Api137, instead, likely hinders passage through the narrow cavity between negatively charged helices H51 and H49 of the 23S rRNA.”

4.4 In Fig. 1, why not test the binding of Api137 and Api88 to the 30S subunit? Could there be additional binding sites in the 30S subunit?

➤ We thank the reviewer for this suggestion and conducted an additional fluorescence polarization assay with purified 30S subunit and the individual peptides (155-58):
 “...Binding to the 30S subunit was approximately 23-fold weaker for Onc112 and Api88 and 10-fold weaker for Api137 compared to the 50S subunit (Fig. 1e), suggesting that the 50S subunit is the major interaction partner of the PrAMPs studied.”

The determined K_d values were added to the updated table in Figure 1:

PrAMP	K_d [$\mu\text{mol/L}$]			
	70S ribosome		50S subunit	30S subunit
	extract	pure		
Api88	0.90 - 1.22 ^[a]	1.82 \pm 0.08	0.60 \pm 0.04	14.67
Api137	0.20 - 0.56 ^[a]	4.73 \pm 0.28	2.15 \pm 0.11	13.54
Onc112	0.03 - 0.09 ^[a]	0.06 \pm 0.005	0.05 \pm 0.001	1.16

4.5 The translation inhibition assays presented in Fig. 1 would benefit from **ribosome toeprinting experiments with Api88 and Api137**, by possibly providing clear evidence as to whether Api88 stalls ribosomes at stop codons or not. This would provide further support to the hypothesis regarding an ‘evolutionary fall-back mechanism’ of inhibition by Api88 put forward by the authors.

We followed the reviewer’s advice and conducted toeprint assays in collaboration with the group of Dr. Shura Mankin (University of Illinois Chicago, USA). Dorothea Klepacki conducted the experiments and appears as co-author now. The toeprint data show that Api88 is only a weak termination inhibitor (lines 123-32):

Fig. 1 d, *In vitro* toeprinting analysis of Api137 and Api88. The start codon is marked by a green and ribosome arrest at the UAG stop codon of the model *yrbA* ORF is marked by a red

arrowhead. The control reaction with no added PrAMPs is labeled as “none”. Sequencing reactions are labeled as C, U, A, G.

4.6 Figures are repetitive with incomplete descriptions. Extended Data Fig. 5 and Fig. 2 are giving the same information.

➤ We thank the reviewer for pointing out this redundancy and have removed repetitive elements from Fig. 2 and **Supp. Fig. 5**, which due to the implementation of new figures is now **Supp. Fig 7**.

4.7 Fig. 3 legend is incomplete. What are the transparent volumes outside the surface charge representations, full volume or volume extractions around the model?

➤ We are grateful to the reviewer for pointing out this inaccuracy and have added the following information to the legend of **Fig. 3d** now **Fig. 4d**:

“**d**, 50S•Api88 conformation III. Coulombic electrostatic potentials are shown in red (electronegative; -10 kcal/mol·e), white (neutral; 0 kcal/mol·e) and blue (electropositive; 10 kcal/mol·e). Surface models are calculated at 4 Å resolution. **For clarity, Api137 and Api88 are highlighted using transparent sphere model in electrostatic representations.**”

4.8 Fig. 4j, is this a rotated view of panel i? Panel h does not seem to be an electrostatic representation.

➤ We thank the reviewer for pointing out a lack of precision in **Fig. 4** (now **Fig. 5**). We have added a small eye in Fig.5 panel i to indicate the respective view in panel j. In addition, correct labeling of panel h was added, mentioning that it is a surface model.

4.9 Fig. 5, please review labels in panels d, e. A1600 is a C, A1348 is a C, A1345 is a C, A1396 is a U. One has to use a very low contour level to visualize “some” density of Api88 at site III. Again, please attempt to better sort particles to improve density.

➤ We thank the reviewer for pointing out the letter mix-up in **Fig. 5** (now **Fig.7**) panels d and e. We have now corrected these bases in the figure.

The masked classification for that binding site resulted in convergence to a single class in every attempt. As mentioned before, we agree that especially the N-terminal region only appears at very low contour level, indicating that the peptide is only partially stabilized in that region. Discriminating between compositional and conformational heterogeneity could result in weaker density and remains challenging to sort for such a small target. Especially concerning the third binding site, we observed a more pronounced density for residues 11-16, which we

highlighted in our manuscript. We agree with reviewer #4 that the density for residues 4-10 is at a very low threshold and assume that these residues are only partially stabilized.

We have added the following details to the text (lines 348-49):

“Although the density appeared weaker compared to binding sites 1 and 2, focused classification did not result in a separation of peptide bound particles.”

Line 351: “The well resolved central region of the peptide is located...”

Line 355: “The weakly resolved N-terminal region...”

4.10 Extended Data Fig. 6a – significance of such a comparison is unclear, considering the dynamic binding mode of Api88.

➤ With this figure panel, we intended to show the differences between the proposed Api137 and tentative Api88 conformations. However, we agree that such a comparison may not be supportive but misleading and have deleted **Supp. Fig. 6a** (now **Supp. Fig. 9a**). In addition, the dynamic binding of Api88 was supported by the new MD simulations (see point 4.5).

4.11 Extended Data Fig. 6g – figure crowded with the unnecessary distances, which are also missing description in the legend.

➤ We agree that the figure appeared crowded due to the use of many colors and redundant distances. We have completed the figure legend, adapted the colors and redesigned the appearance of **Supp. Fig. 6** (now **Supp. Fig. 9**).

4.12 Extended Data Fig. 10 – legend unclear, needs more and better description.

➤ We have updated Supp. Fig. 10 (now **Supp. Fig. 13**) and added a more detailed description within the figure legend.

Overall peptide geometry at different binding sites. Canonical conformation of Api137 within the PET is shown in grey for comparison. N-terminal (dark red), central (blue) and C-terminal (light red) sections of modeled residues at different sites were rigid body docked into a 4 Å molmap of Api137. Residues used for docking are labeled using respective color and are shown as sticks **a**, Modeled residues of Api137 at the tunnel end. **b**, Modeled residues of Api88 conformation I within the PET. **c**, Modeled residues of Api88 conformation II within the PET. **d**, Modeled residues of Api88 conformation III within the PET. **e**, Modeled residues of Api88 at the third binding site.

4.13 In several instances, the authors are missing important citations. For example, please cite the **Steitz** group as they also published structures of Onc112 and Bac7 bound to the ribosome (**NSMB 2015, NAR 2016**). The studies were published back-to-back together with the Innis group in the same journals.

➤ We are grateful to the reviewer for suggesting these relevant publications that unintendedly escaped our attention. We added the citations as follows (lines 236-39):

“A similar tendency is observed when compared with other PrAMPs (Supplementary Fig. 9b). Onc112 [9, 26], bactenecin-7 (Bac7(1-16)) [27, 28], pyrrolicorin (Pyr) [27, 28], metalnikowin I (Met) [27, 28] and drosocin (Dro) [29] explore a broad area...”

And lines 422-24: “Remarkably, all conformations of Api88 were found to be more similar to Onc112 [9, 26] and Bac7(1-16) [27, 28] than to Api137 ..”

In addition, we have added the citations in Supp. Fig. 9:

“Supplementary Fig. 9: Comparison of known PrAMP conformations in the PTC including Api137 and Api88. a, Conformational space occupied by conformations I (pale orange), II (orange), III (light red) of Api88 and Api137 (yellow) within the tunnel binding site. Distances between selected C α atoms are indicated using blue dots and dashed lines. Distances are color-coded (<4 Å = light blue, 4-6 Å cornflower blue, >6 Å dark blue. **b,** Conformational space occupied by Api88, Api137, and other PrAMPs within the tunnel (Onc112 (PDB: 4ZER[9,26], Met (PDB: 5FDU[27,28], Pyr (PDB: 5FDV[26,27], Bac7(1-16) (PDB: 5F8K[27,28], Dro1 (PDB: 8ANA[29]. Distances between selected C α atoms are indicated using blue dots and dashed lines. Distances are color-coded (<4 Å = light blue, 4-6 Å cornflower blue, >6 Å dark blue. **c-e,** Comparison of **c,** Api137, **d,** Api88 conformation I, and **e,** Api88 conformation III with the position and conformation of Onc112). **f,** Comparison of Api88 conformation III with the position and conformation of Bac7(1-16; brown). **g,** Residues of Api137(Y7), Api88(Y7), and Dro(R9) interacting with A751 of the 23S rRNA.”

4.14 Line 47: “ESKAPE” and not “ESKAPEE”.

➤ It turns out that both abbreviations are correct. The latter one being even more up to date since it includes pathogenic *E. coli*. For reference see Yu et al., 2020: <https://doi.org/10.1016/j.fitote.2019.104433>. Therefore, we keep the abbreviation “ESKAPEE”.

Enterococcus faecium, Staphylococcus aureus, Klebsiella pneumoniae, Acinetobacter baumannii, Pseudomonas aeruginosa, Enterobacter spp., Escherichia coli

4.15 Line 75: “bypass” would be better with “readthrough”

➤ We changed the wording accordingly (line 77).

4.16 Lines 92-94: “First, we established a protocol for the isolation of high-purity *E. coli* 70S ribosomes [17] uncontaminated by other proteins, including the release factors RF1 and RF2 (Extended Data Fig. 1),...” “we” is not clear as none of the author seem to have participated in the work referenced (ref. 17). Also, how is this purification procedure better to remove contaminant proteins? It is not clear from the Supp. Fig. 1.

➤ The reviewer's confusion shows us that the issue needs to be presented more clearly. It was our intention to purify ribosomes free of additional factors (non-core ribosomal proteins) such as RF1 and RF2. We changed the text accordingly (lines 134-39):

“In addition to the bacterial ribosome as the main target, Api137 and Api88 also bind to other proteins, such as the heat shock protein DnaK and ribosome-associated release factors [5,9]. These and other proteins present in previously used ribosome preparations [11,15,21] will most likely affect the measured dissociation constant (K_d). Therefore, we used an additional chromatographic purification to obtain highly pure *E. coli* 70S ribosomes (Supp. Fig. 1) [22] and thus more reliable 70S ribosome binding constants.”

Also, the caption of Supp. Fig. 1 was modified:

“b, Coomassie Brilliant Blue-stained gel of an SDS-PAGE (T=16%) obtained by analyzing the 70S purification steps. First, ribosomal extract containing other bacterial proteins, which was used for further purification by anion exchange chromatography and the received fractions obtained by AEC (panel A), divided into flow through, wash, showing both separated bacterial proteins from 70S ribosome, the elution of 70S without other proteins, the concentrated pure *E. coli* 70S ribosome...”

4.17 Line 189: Supp. Fig. 5f should be Ext Data Fig. 6f

We have corrected the labeling of Supp. Fig. 5f (now Supp. Fig. 9f), which we redesigned to increase clarity. See also reviewer#2 point 6.

4.18 Lines 208-218: refer to specific panels in Ext. Data Fig. 8

➤ As suggested, we refer to specific panels of the former Supp. Fig. 8, now Supp. Fig. 11 (lines 299-305):

“The Api137 structures within vacant 50S and 70S in the presence of RF1 [9] are geometrically highly compatible (Supplementary Fig. 11a, b, e). In the absence of release factor, Arg17 formed a hydrogen bond to the backbone of G2505. In the presence of RF1, the C-terminus shifted slightly into the tunnel. In conformations I and II of Api88, Arg17 is involved in a stacking interaction with U2504, leaving the side chain in the correct position (Supplementary Fig. 11c & f). In contrast, conformation III of Api88 is rotated and shifted compared to Api137 (Supplementary Fig. 11d & g).”

4.19 Lines 322 and 324: π , use the Greek letter instead

➤ As suggested, we used the Greek letter. Due to reorganization of the discussion, we now refer to π interactions in lines 402-3 and 410-11:

“They contribute to the positive net charge, and they can participate in π -interactions with nucleobases.”

“Following the electric charge, the peptides move along the nucleobase rugs, π -interacting with the amino acid side chains.”

4.20 Line 328: “...via distinct binding sites the trigger factor (TF)...” relative to the trigger factor?

➤ We changed the wording as follows (lines 445-48):

“These ribosomal proteins are located adjacent to the PET exit and are binding hubs for the molecular chaperone trigger factor (TF) [34] and the signal recognition particle (SRP)...”

4.21 There are three corresponding authors. Yet, only two email addresses are provided.

➤ We thank the reviewer for this hint and have added the contact details of the third corresponding author Rainer Nikolay.

4.22 On the front page, the star refers to a “present address”. However, there is no star in the authors’ list.

➤ Once again, we thank the reviewer for the hint and have listed both the current and former address of Rainer Nikolay (⁷Max Planck Institute for Molecular Genetics, Department of Genome Regulation ... ¹Institute of Medical Physics and Biophysics, Charité ...).

We thank reviewer#4 for the critical and constructive review.

Reviewer #5 (Remarks to the Author):

Antibiotic resistance is a global phenomenon exacerbated in hospital settings, resulting in increased patient morbidity rates. Development of novel therapeutics that overcome antibiotic resistance are therefore important to circumvent this challenge. Api137 and the C-terminal amine-modified Api88 are peptide therapeutics that have been demonstrated previously to be effective antibiotics, particularly targeting, and inhibiting proper mRNA translation by the 70S ribosome. Lauer S, et al present a combination of translation inhibition, binding affinity, cryo-EM, and mass spectrometric data to better understand the differences in binding of Api137 and Api88 to the 70S and 50S E. coli ribosomal subunits. Using this combinatorial approach, they were able to show differences in the translation inhibition efficiency, binding affinity, and binding modes between Api137 and Api88 and conclude that these differences may explain the differences in their mechanism of action. Understanding the structure-function relationship when designing therapeutics is critical for informed drug design and this work is a valuable contribution to this effort.

Overall, I recommend this paper to be published. Prior to publication, I do have some edits that I believe would help to further improve the clarity and quality of the work presented, which I outline below. Please note that since I am not a cryo-EM expert, I am not able to comment on the validity of the data acquisition, processing, and interpretation presented here, and trust that other reviewers will make sound recommendations based on their expertise in this area.

We appreciate that reviewer#5 recommends our paper to be published.

PS: We noticed that, according to Nature Communications formatting requirements, the correct term for figures in the supplement is **not** “Extended Data Fig.” **but rather** “Supplementary Fig.”. In the revised version of the manuscript, we use the term “Supplementary Fig.”.

5.1. Due to the broad research audience of Nature Communications articles, improvement of clarity of the text to enhance the readers comprehension of the overall significance of the work would be beneficial. For example, many readers may not be experts in AMPs and therefore may not understand why “AMPs attacking bacterial membranes often have low safety margins”. Please clarify this in the text with a few words or one sentence to explain this. The overall text is quite well written, and just requires a few minor tweaks like the one described above for minor improvement.

➤ We fully agree and shortly explained and clarified why these AMPs have low safety margins. Lines 57-62: “However, AMPs attacking bacterial membranes often have low safety margins, which does not allow them to be administered at pharmacologically desired high doses to maintain a minimum blood concentration for longer periods of time before the next injection. Therefore, such lytic peptides are typically considered for topical applications. AMPs that specifically inhibit intracellular bacterial targets are more promising lead compounds for systemic treatments.”

5.2. The dissociation constants presented should be stressed by the authors to be apparent Kd values since they report 2 binding sites for Api137 and 3 for Api88. With the technique used to measure binding affinity, it is not possible to parse between the binding events and thus these reported Kd values represent an apparent value for a combination of these binding events taking

place. This could explain the discrepancy between Api88 having a lower K_d , but being less effective at inhibiting translation.

➤ We agree and therefore added the following statement (lines 387-94): “The different binding modes were further supported by the 10- to 20-fold higher K_d values obtained for Api137 after purification of ribosomal extracts to remove release factors and other contaminating proteins, whereas the K_d value of Api88 remained rather stable, similar to Onc112. In addition, high resolution cryo-EM reconstructions of purified 50S subunits incubated with Api137 or Api88 confirmed multiple binding sites for both peptides. In this respect the reported K_d values should be considered as apparent K_d values, as they represent the average K_d of at least two or three binding sites of Api137 and Api88, respectively, although there may be additional binding sites not visible in the structures.”

5.3. The authors discuss MS data in the main text, have several methods sections about it, but the data is not presented anywhere in either the main text figures or supplemental figures. Inclusion in supplemental material seems appropriate, and making raw and processed files openly available could also be completed for data transparency purposes and is highly encouraged in the mass spec community (can upload data to MassIVE data repository at <https://massive.ucsd.edu/ProteoSAFe/static/massive.jsp> and make the data openly available once the manuscript is published). Below is the content I request the authors to address:

➤ We uploaded the MS data to panorama with the ProteomeXchange ID PXD044892. The DOI is <https://doi.org/10.6069/3f7w-2t74>. The permanent link of the unique identifier is <https://panoramaweb.org/Ribosome.url>, which will be publicly available after the manuscript has been accepted.

5.3.i) Inclusion of the MALDI-TOF-MS and ESI-MS spectra that were used to confirm the intact masses of each peptide species seems appropriate to include in the supplemental material since this information was used to confirm correct peptide synthesis. This should be included for all **Api137 and Api88 species**, including the **modified Api88** used for the crosslinking experiments.

➤ We thank the reviewer for pointing out the need to share the MS spectra of the Api peptides for data transparency purposes. All mass spectra and chromatograms obtained for peptides Api88, Api137, and Onc112 are shown in Supp. Fig. 14 and further modified peptides used in this study in Supp. Fig. 15.

5.3.ii) The crosslinked in-gel digestion experiment is only summarized in a table in supplemental in Extended Data Fig 3 c, where the authors present summed peak area for the target proteins. The Western blot makes it clear that some protein species are enriched in the Api88 treated samples compared to controls, but the limited information given in the table does not convincingly suggest the identity of these proteins to be primarily L10, L23, and L29. Identifying the most abundant protein species in each dataset would be valuable, and if comparison between the control vs Api88 treated samples is desired, then comparing the peak area for a particular peptide, rather than the summed peak area, would be more informative. In this case, addition of an example chromatographic peak with overlapping charge states and complementary MS/MS spectrum for the peptide of interest would add useful information, while having access to the raw and processed data files would make it possible for interested readers to analyze the data more deeply if desired. Protein sequence coverage is also something

that could be added to the table to help the reader understand the validity of the protein assignments.

➤ The MS data were uploaded to panorama (See response to critique 5.3.i) and additional information is provided in the supplement:

The most abundant proteins identified by mass spectrometry of the pooled sample are listed in Supp. Fig. 5. Two additional figures were added: Ext, Data Fig. 4 visualizes the difference between non-enriched and enriched peptides, corresponding to the summed-up peak areas introduced in Fig. 3c. Moreover, the peptides of an unspecified-enriched protein are shown in Supp. Fig. 5. The protein sequence coverages of the enriched proteins are shown in Supp. Fig. 3c.

5.3.iii) Related to the above comment, I wonder if the **crosslinked Api88 to peptides originating from the L10, L23, and L29 proteins** were found in the data? This would be very convincing that these are the proteins in contact with Api88 and would eliminate much of the ambiguity in the results described in ii). If this is not feasible, **please describe why you only search for the non-crosslinked peptides in your analysis within the body of your text** (page 5 lines 125-131).

➤ This is an important issue that was also raised by reviewer#2 (point 3). Indeed, our goal was to detect the crosslinked Api88 to peptides originating from the uL10, uL23, and uL29 proteins and thereby identify the interaction sites between Api88 and the ribosomal proteins. However, despite significant efforts (use of different cross-link software's, investigations by marker ions and additionally use of isotope labelled peptides (all data not shown), it was not possible to detect these crosslinked peptides.

We added the following text to the section **UV-activatable Api88 forms cross-links with ribosomal proteins** that was reorganized upon suggestion by multiple reviewers (lines 178-80): “Despite significant efforts, we failed to detect cross-linked complexes consisting of Api88 and ribosomal proteins by MS analysis directly. Therefore, proteins cross-linked to biotin-SG-Api88(Y7B) were enriched ...”

We thank reviewer#5 for the critical and constructive review.

REVIEWERS' COMMENTS

Reviewer #1 (Remarks to the Author):

The authors resolved all the concerns and questions. I recommend the work for publication.

Reviewer #3 (Remarks to the Author):

The reviewed manuscript by Lauer et al has been significantly altered and improved.

The authors have answered effectively and convincingly a set of questions posed by 4 referees, which has led to an extensive rewriting of the manuscript (including title and abstract), adding significant new data and making it more effective.

In this respect, have no further questions or comments.

Reviewer #4 (Remarks to the Author):

I commend the authors for seriously considering my suggestions and performing additional experiments to address my initial concerns (toeprinting, focused particle classification, MD simulations, binding assays to free 30S subunits). The manuscript is much improved and clearer to the reader. I recommend publication after the following minor issues are addressed:

Line 73: "...blocking the PET and stalling protein translation at the initiation of translation [9]."

For the sake of accuracy, the authors should also refer to the work from the Steitz group here by adding reference 26. The two crystal structures of 70S ribosomes bound to Onc112 were published back-to-back with the Innis group in 2015.

Line 131: "...about to release the mRNA at the stop codon." Is mRNA correct here or do the authors mean nascent peptide chain?

Line 337: "...ribosomal rRNA..." redundant, ribosomal RNA

Line 343: Missing reference Roy RN et al. NSMB 2015.

Same at line 381

Lines 378-379: "Our reports identifying the bacterial ribosome as a major target of PrAMPs and localizing their binding site to the nascent PET [7,8]..." The first part of the sentence is accurate. However, this reviewer is not aware that the binding to the nascent PET was identified before the structural studies. Therefore, the second part of this sentence is inaccurate – needs rephrasing.

Lines 404-405: "Upon release of the nascent chain and following diffusion or the negative net charge of the PET interior..." the sentence is unclear. Is "or" appropriate here?

Typo Line 410 - 'rungs' for rugs

Legend of Fig. 6, line 94:

"...Bac7(1-16) (PDB: 5F8K [9,26]),..." Wrong references. Should refer to Bac7 structure papers, references 26,27.

Legend of Fig. 8, line 110:

"...associate with the negatively charged rRNA of the 70S subunit." 70S ribosome

Finally, the authors should pay attention to grammatical errors:

For example:

Line 58: "...which does not allow them to be administered..." which impede their administration?

Lines 76-77: "It inhibits translation termination by trapping the release factors RF1 and RF2, which arrests translating ribosomes at the stop codon or promotes..." arrest...promote...

Line 407: "...an array of exposed nucleobases protrudes into..." protrude

Reviewer #5 (Remarks to the Author):

After reading the response to reviewers document and the updated manuscript file, my recommendation is to publish this article. All of the points raised by myself and others have been addressed appropriately in the revised manuscript.

Reviewer #6 (Remarks to the Author):

The authors use cryo-EM to examine the mechanism of action of peptide antibiotics that target the ribosome. Specifically, they focus on Api137 and Api88, which bind near the polypeptide exit tunnel adjacent to the peptidyl transferase center. They find that both drugs have alternative binding sites that aid in repressing translation and conclude that there may be multiple mechanisms for each drug. In light of the crisis of antibiotic resistance that permeates US hospitals, the search for new, effective antibiotics is of critical interest for biomedical research worldwide. As the current manuscript addresses this problem directly, it is highly relevant and of great import. In particular, peptide-based antibiotics represent one of the most promising alternative antibiotic classes. Since these are less well-studied, research in this area is highly sought after. The authors have shown an important trend of multimodel mechanisms that may be key to understanding this antibiotic class. The peptide synthesis, ribosome preparation, toe print biochemical assays, cryo-EM and molecular dynamics simulations appear to have been done correctly. Performing 5 simulations of 2 microseconds each (10 microseconds aggregate sampling per case) is sufficient to assess the ensemble of conformations sampled inside the ribosome tunnel, as the tunnel itself is a dramatic constraint on the conformation of the peptide. Overall, this study is an important contribution to the field and should be published without delay.

REVIEWER COMMENTS 2

Reviewer #1 (Remarks to the Author):

The authors resolved all the concerns and questions. I recommend the work for publication.

- We thank reviewer #1 for recommending this manuscript for publication.

Reviewer #2 (Remarks to the Author):

Reviewer #3 (Remarks to the Author):

The reviewed manuscript by Lauer et al has been significantly altered and improved.

The authors have answered effectively and convincingly a set of questions posed by 4 referees, which has led to an extensive rewriting of the manuscript (including title and abstract), adding significant new data and making it more effective.

In this respect, have no further questions or comments.

- We thank reviewer #3 for the positive feedback and the recommendation to publish this manuscript.

Reviewer #4 (Remarks to the Author):

I commend the authors for seriously considering my suggestions and performing additional experiments to address my initial concerns (toeprinting, focused particle classification, MD simulations, binding assays to free 30S subunits). The manuscript is much improved and clearer to the reader. I recommend publication after the following minor issues are addressed:

- We thank reviewer #4 for appreciating our efforts on the newly added experiments, the careful review, and the recommendation to publish this manuscript.

Line 73: "...blocking the PET and stalling protein translation at the initiation of translation [9]." **(Reference)**

For the sake of accuracy, the authors should also refer to the work from the Steitz group here by adding reference 26. The two crystal structures of 70S ribosomes bound to Onc112 were published back-to-back with the Innis group in 2015.

- We added the reference.

Line 131: "...about to release the mRNA at the stop codon." Is mRNA correct here or do the authors mean nascent peptide chain?

- We have corrected it accordingly and changed "...about to release the nascent peptide chain at the stop codon."

Line 337: "...ribosomal rRNA..." redundant, ribosomal RNA

- Line 335. We have corrected it accordingly and removed "ribosomal" ending up in "... and the rRNA."

Line 343: Missing reference Roy RN et al. NSMB 2015. **(Reference)**

- We have added the reference.

Same at line 381 **(Reference)**

- We have added the reference.

Lines 378-379: "Our reports identifying the bacterial ribosome as a major target of PrAMPs and localizing their binding site to the nascent PET [7,8]..." The first part of the sentence is accurate. However, this reviewer is not aware that the binding to the nascent PET was identified before the structural studies. Therefore, the second part of this sentence is inaccurate – needs rephrasing.

- In [8], competitive binding assays with common antibiotics such as erythromycin or chloramphenicol revealed an overlapping interaction site within the PET, resulting in the conclusion that Api137 binds within the PET. Therefore, we clarified in line 379 – 380, "Our reports identifying the bacterial ribosome as a major target of PrAMPs and **mapping** their binding site to the nascent PET [7,8]..."

Lines 404-405: "Upon release of the nascent chain and following diffusion or the negative net charge of the PET interior..." the sentence is unclear. Is "or" appropriate here?

- Line 405 - 406. We have corrected it accordingly "...release of the nascent chain and following diffusion as well as the negative net charge of the PET interior..."

Typo Line 410 - 'rungs' for rugs

- Changed accordingly to “rungs”

Legend of Fig. 6, line 94:

“...Bac7(1-16) (PDB: 5F8K [9,26]),...” Wrong references. Should refer to Bac7 structure papers, references 26,27. **(Reference)**

- We now cite the following references:

[27] Gagnon, M. G. *et al.* Structures of proline-rich peptides bound to the ribosome reveal a common mechanism of protein synthesis inhibition. *Nucleic Acids Res* **44**, 2439–2450 (2016).

[28] Seefeldt, A. C. *et al.* Structure of the mammalian antimicrobial peptide Bac7(1–16) bound within the exit tunnel of a bacterial ribosome. *Nucleic Acids Res* **44**, 2429–2438 (2016).

Legend of Fig. 8, line 110:

“...associate with the negatively charged rRNA of the 70S subunit.” 70S ribosome

- We have corrected it accordingly to “70S ribosome”.

Finally, the authors should pay attention to grammatical errors:

For example:

Line 58: “...which does not allow them to be administered...” which impede their administration?

- We have corrected it accordingly and changed “...which does not allow them to be applied at pharmacologically...”

Lines 76-77: “It inhibits translation termination by trapping the release factors RF1 and RF2, which arrests translating ribosomes at the stop codon or promotes...”
arrest...promote...

- We have corrected it accordingly and changed “It inhibits translation termination by trapping the release factors RF1 and RF2, which arrest translating ribosomes at the stop codon or promote...”

Line 407: “...an array of exposed nucleobases protrudes into...” protrude

- We note that "protrudes" is indeed correct, and therefore, we will leave it unchanged.

Reviewer #5 (Remarks to the Author):

After reading the response to reviewers document and the updated manuscript file, my recommendation is to publish this article. All of the points raised by myself and others have been addressed appropriately in the revised manuscript.

- We thank reviewer #5 for recommending this manuscript for publication.

Reviewer #6 (Remarks to the Author):

The authors use cryo-EM to examine the mechanism of action of peptide antibiotics that target the ribosome. Specifically, they focus on Api137 and Api88, which bind near the polypeptide exit tunnel adjacent to the peptidyl transferase center. They find that both drugs have alternative binding sites that aid in repressing translation and conclude that there may be multiple mechanisms for each drug. In light of the crisis of antibiotic resistance that permeates US hospitals, the search for new, effective antibiotics is of critical interest for biomedical research worldwide. As the current manuscript addresses this problem directly, it is highly relevant and of great import. In particular, peptide-based antibiotics represent one of the most promising alternative antibiotic classes. Since these are less well-studied, research in this area is highly sought after. The authors have shown an important trend of multimodel mechanisms that may be key to understanding this antibiotic class. The peptide synthesis, ribosome preparation, toe print biochemical assays, cryo-EM and molecular dynamics simulations appear to have been done correctly. Performing 5 simulations of 2 microseconds each (10 microseconds aggregate sampling per case) is sufficient to assess the ensemble of conformations sampled inside the ribosome tunnel, as the tunnel itself is a dramatic constraint on the conformation of the peptide. Overall, this study is an important contribution to the field and should be published without delay.

- We express our gratitude to reviewer #6 for highlighting the significance of our research topic and recommending this manuscript for publication.